# Beyond Single-Task: Robust Multi-Task Length Generalization for LLMs

Yi Hu[1,*]    Shijia Kang[1,*]    Haotong Yang[1]    Haotian Xu[2]    Muhan Zhang[1,†]

[1]Institute for Artificial Intelligence, Peking University

[2]Xiaohongshu Inc.

## Abstract

Length generalization—the ability to solve problems longer than those seen during training—remains a critical challenge for large language models (LLMs). Previous work modifies positional encodings (PEs) and data formats to improve length generalization on specific symbolic tasks such as addition and sorting. However, these approaches are fundamentally limited to special tasks, often degrading general language performance. Furthermore, they are typically evaluated on small transformers trained from scratch on single tasks and can cause performance drop when applied during post-training stage of practical LLMs with general capabilities. Hu et al. [19] proposed Rule-Following Fine-Tuning (RFFT) to improve length generalization in the post-training stage of LLMs. Despite its compatibility with practical models and strong performance, RFFT is proposed for single tasks too, requiring re-training for each individual task with extensive examples. In this paper, we study length generalization in multi-task settings and propose *Meta Rule-Following Fine-Tuning (Meta-RFFT)*, the first framework enabling robust *cross-task* length generalization. As our first contribution, we construct a large length generalization dataset containing **86 tasks** spanning code execution, number processing, symbolic and logical reasoning, beyond the common addition or multiplication tasks. Secondly, we show that cross-task length generalization is possible with Meta-RFFT—after training on a large number of tasks and instances, the models achieve remarkable length generalization ability on *unseen* tasks with *minimal fine-tuning or one-shot prompting*. For example, after fine-tuning on 1 to 5 digit addition, our 32B model **achieves 95% accuracy on 30 digit addition**, significantly outperforming the state-of-the-art reasoning models (DeepSeek-R1-671B: 72%; QwQ-32B: 32%), despite never seeing this task during RF-pretraining. Our code is available at https://github.com/MuLabPKU/Meta-RFFT.

.

## 1 Introduction

Large language models (LLMs) have achieved revolutionary performance in a wide range of tasks, from natural language understanding and generation to complex reasoning [33, 34, 16, 38, 9, 8]. However, they still face challenges when processing some basic tasks seemingly intuitive to humans. One of the challenging problems is the **length generalization**, where Nogueira et al. [31], Zhou et al. [58, 60], Anil et al. [1] reveal that transformers suffer from a significant performance drop when solution steps exceed the training range, suggesting that models fail to capture the inner mechanism in these reasoning problems. A classic example is long-integer addition: models trained on addition problems with fewer digits often fail to generalize to higher-digit cases.

---

*Equal contribution. † Corresponding author. Muhan Zhang <muhan@pku.edu.cn>

39th Conference on Neural Information Processing Systems (NeurIPS 2025).

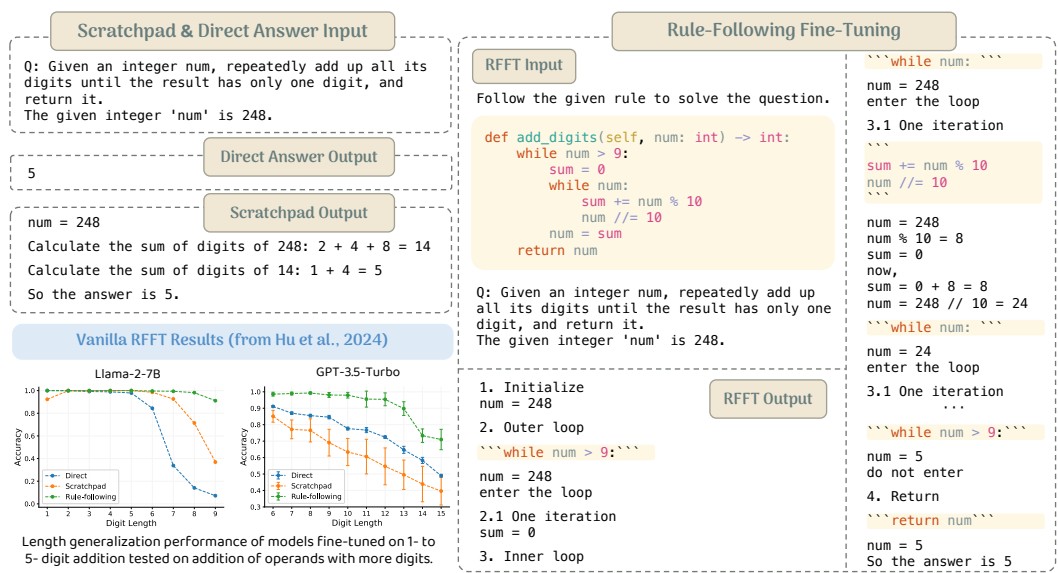

Figure 1: Comparison of input-output sequences across three methods: *direct answer*, *scratchpad* (top left), and *RFFT* (right), with single-task performance results shown at the bottom left.

Although long chain-of-thought (CoT) models seem to learn a plausible reasoning process for some complex tasks such as math / code [35, 10, 44, 45, 56, 51], length generalization is still a challenge for them. Even advanced long-CoT models like DeepSeek-R1-671B and QwQ-32B exhibit unsatisfactory performance in long integer addition, achieving only 72% and 32% accuracy, respectively, on 30-digit problems.

Some prior work improves length generalization by designing specialized positional encodings (PEs) or data formats. [58, 41, 21, 23, 60, 4, 28, 5, 2]. However, these approaches are heavily dependent on specific properties of the objects (like numbers) and thus limited to specialized domains. Besides, these methods are shown to be effective through training small transformers from scratch on a single task, yet proven ineffective for fine-tuning on top of pretrained LLMs [52], fundamentally due to their incompatibility with the PEs / number formats used for the general corpus. This greatly limits their practical applicability.

Regarding enhancing length generalization in the post-training stage of LLMs, Anil et al. [1] states that direct answer and scratchpad fine-tuning [32] (examples are shown in Figure 1) are not enough to enable robust length generalization. Recently, Hu et al. [19] found that the issue stems from the case-based mechanism of LLM reasoning and proposed **R**ule-**F**ollowing **F**ine-**T**uning (RFFT) to teach models to follow rules step by step. RFFT represents the first successful effort to solve diverse length generalization tasks using a unified approach. As illustrated in Figure 1, RFFT explicitly incorporates rules into the input, guiding the model to follow them strictly. In contrast, the baseline scratchpad fine-tuning method only provides intermediate computations without conveying the underlying rules, similar to teaching children addition solely through examples, without explaining the principles. By explicitly instructing the model in both the rules and their execution traces, RFFT significantly improves length generalization: GPT-3.5-Turbo fine-tuned on 1-5 digit addition achieves **over 95% accuracy on 12-digit addition**, surpassing scratchpad fine-tuning **by 40%**.

Despite the compatibility with general tasks and pretrained LLMs, RFFT [19] is proposed for *single-task settings*, where they prepare data and fine-tune the models on three tasks separately: addition, base-9 addition and last letter concatenation. This setting is impractical for users and limits model generalizability. Users must prepare task-specific rule-following datasets and perform extensive fine-tuning, which is a costly process that requires separate models for each task. More critically, single-task RFFT only *models the relationship between one specific rule and its instances*, failing to *leverage the shared structures and generalization potential across different rules*.

In this paper, we investigate the task transferability and generalization of the rule following capacity of models, and propose **Meta R**ule-**F**ollowing **F**ine-**T**uning (Meta-RFFT). We find that through fine-

tuning on a large-scale rule-following dataset with diverse tasks, a model shows positive transferability on *unseen* tasks with minimal adaptation.

As our first contribution, we **collect 86 different length generalization tasks with 310k training samples** from four task domains including code execution, number processing, logic and symbolic reasoning, which significantly broadens the previous length generalization tasks which mainly focus on addition, sorting or other basic operations. For each task, we manually annotate the code (or pseudo-code) for each task as its rule, as well as a template script that can generate a detailed trajectory process for rule-following for each question. Based on these template scripts, by simply providing the problem variables, the corresponding rule-following trajectory at any desired length can be automatically generated, which can then be used to train models. Finally, we collect 310k training samples on 74 tasks while the other 12 tasks are reserved as test sets.

In the experiments, the models are first fine-tuned on 74 RFFT tasks (we call it as rule-following *"pretraining"*), leading to a rule-following foundation model. Then, the models are further adapted to the downstream task with minimal fine-tuning. These two-stage models show significantly better performance than both baseline models (like direct answer or CoT of reasoning models) and the single-task RFFT models. Specifically, **a 32B model fine-tuned on 1-5-digit addition** achieves **95% accuracy on 30-digit addition**, vastly outperforming reasoning models of comparable or much larger parameter size (**DeepSeek-R1-671B: 72%; QwQ-32B: 32%**) and **vanilla RFFT (40%)**.

Notably, these models with rule-following pretraining can solve unseen tasks with high accuracy with the help of **only one example**, suggesting these models learn a task-generalized **in-context rule-following ability**. This capacity means these models can be directly used by users who cannot modify model parameters, as long as one example to exemplify the rules is provided in the context. At the same time, the model can also generalize to rules written in natural language style.

We further demonstrate that the foundation model robustly **acquires shared computational primitives** (e.g., loop maintenance), which are critical for cross-task generalization. Our experiments reveal that in vanilla RFFT, where models are trained separately for each task, loop maintenance is a primary error source. In contrast, Meta-RFFT, where models are enhanced by rule-following pretraining on tasks with shared computational structures, exhibits significantly more precise loop maintenance in downstream tasks. These findings confirm that meta-rule-following capability stems from mastering transferable computational patterns rather than task-specific ones.

In summary, we construct a large-scale length generalization dataset comprising 86 diverse tasks spanning diverse domains, enabling systematic study of rule-following transferability (§3.2). Our proposed Meta-RFFT framework demonstrates that multi-task post-training on 74 tasks facilitates strong length generalization on unseen tasks with minimal downstream fine-tuning (§4.2) or even 1-shot prompting (§4.3). Crucially, we identify that this transferability stems from models learning shared computational primitives that underlie diverse tasks (§4.2), while maintaining robust performance when rule formats transition from formal code to natural language (§5).

## 2  Related Work

**Length generalization.**   A series of studies have attempted to tackle this issue by modifying positional encodings (PEs) and data formats [58, 41, 21, 23, 60, 4, 28, 5]. However, these efforts face several key limitations. First, the proposed PEs and data formats are often specifically tailored to symbolic tasks, making them difficult to generalize to broader tasks. Second, the methods are typically tested on small-scale models trained from scratch and do not scale well to practical-scale LLMs. Another research direction, including single-task RFFT [19, 17, 52], addresses length generalization by training models on explicit rules and more elaborate reasoning processes.

**Case-based reasoning or rule-based reasoning.**   A central question in understanding LLM reasoning is whether their strong performance stems from pattern matching or mere memorization (or "case-based reasoning" in  Hu et al. [19]), or genuine rule acquisition. Recent studies reveal that LLMs often rely on memorized examples and shortcuts rather than systematic reasoning. Studies show they struggle with counterfactual reasoning [49, 54], reason via subgraph matching [13], and depend on nearby examples for math tasks rather than general rules [19]. On the other hand, research on "grokking" [37, 26, 30, 57] suggests that models can learn interpretable rules of arithmetic reasoning long after overfitting the training set. Yet this phenomenon remains limited to single-task settings. It

is unclear whether rule-based reasoning scales to multitask LLMs. To bridge this gap, we propose Meta-RFFT, which trains models to follow explicit rules across diverse tasks.

**Instruction following.** Following natural language instructions is a fundamental capability of LLMs. Prior work has introduced numerous datasets to enhance this ability [22, 43, 50, 47], enabling models to leverage their underlying knowledge, interact naturally with users [36], and handle diverse tasks [55]. However, existing models trained on these datasets still struggle to accurately execute complex instructions, primarily due to limited high-quality training data [11]. Consequently, ensuring strict adherence to complex instructions—or in this paper, precisely following rules to achieve robust length generalization—remains an open challenge. In our experiments, we use instruction-tuned LLMs (e.g., Qwen-2.5-7B-Instruct and Qwen-2.5-32B-Instruct [39]) as baselines. We further compare Meta-RFFT with standard instruction-following fine-tuning on downstream tasks, demonstrating Meta-RFFT's superior performance.

**LLMs with programs.** Numerous efforts have been made to integrate programs with LLMs to enhance the capabilities of both. LLMs can help with code execution [24] and help developers write code and debug more efficiently [20, 53, 46]. Besides, the formal structure of code helps with task decomposition and enhances LLM reasoning [15, 3, 18, 24, 32].

# 3 Methods

## 3.1 Meta Rule-Following Fine-Tuning (Meta-RFFT)

In this section, we first review vanilla RFFT proposed by Hu et al. [19], and introduce Meta-RFFT.

**Vanilla RFFT** Hu et al. [19] proposes RFFT, which, as shown in Figure 1, includes the rules to solve the task in the input and detailed rule-execution process in the output. We refer to the method from Hu et al. [19] as vanilla RFFT in this paper and identify three key components as follows: (1) the *rules* required to solve a task must be *explicitly provided in the input*; (2) before executing an action, the model is required to *recite the corresponding rule* to ensure precise adherence; and (3) the model must *describe the variables modified by the action*, detailing their states before and after execution. While we primarily use programmatic representations of rules to ensure clarity and precision, the rules discussed in this paper are not limited to code. In Section 5, we also explore natural language rules and investigate the model's ability to transfer rule formats from code to natural language.

**Meta-RFFT** As illustrated in Figure 2, Meta-RFFT adopts a two-stage pipeline:

**1) RF-Pretraining:** We first *fine-tune* the model on a diverse set of rule-following tasks of lengths of 1-15. The tasks span code execution, number processing, symbolic and logic reasoning, which are detailed in Section 3.2. It should be noted that RF-pretraining is a *supervised fine-tuning* process on a large-scale dataset.

**2) Downstream Adaptation:** The model is then adapted to target tasks via either (i) minimal fine-tuning on 1-5-length samples or (ii) 1-shot prompting using exemplars of fewer than 5 steps. Crucially, evaluation is performed on tasks *unseen* during RF-pretraining to assess cross-task and even cross-domain transferability.

During RF-pretraining, the model is expected to grasp the shared commonalities and fundamental rule concepts, such as loop. By leveraging these shared structures, Meta-RFFT enables models to adapt to new target tasks with minimal fine-tuning or even solve them through few-shot

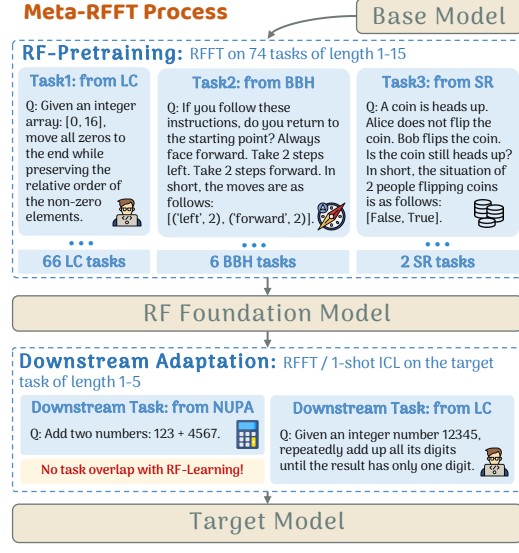

Figure 2: The pipeline of *Meta-RFFT*. *LC*, *BBH*, *SR* and *NUPA* stands for LeetCode, Big-Bench Hard [42], Symbolic Reasoning [48] and the NUPA Benchmark [52] respectively.

prompts. Additionally, training across multiple tasks relieves overfitting the task-specific patterns, encouraging the model to transform from case-based reasoning to more robust rule-based reasoning.

## 3.2 Data Construction

To extend the horizon of length generalization, and to facilitate large-scale multi-task training as well as comprehensive evaluation, we find it essential to construct a large-scale length generalization dataset. When selecting these tasks, we follow these guiding principles:

**1) Tasks must inherently require length generalization.** Specifically, solving the tasks should depend on iterative reasoning. For example, the coin flip task shown in Figure 2 necessitates enumerating each participant's actions to determine the final state of the coin. Here, we use the number of participants to denote "length".

Table 1: The statistics of our dataset. We list the number of tasks collected from each data source and their corresponding split in the RF-pretraining stage or the downstream adaptation stage.

| Data Source | RF-Pretrain | Adaptation | Total |
|---|---|---|---|
| LeetCode | 66 | 8 | 74 |
| NUPA | 0 | 4 | 4 |
| Big-Bench Hard | 6 | 0 | 6 |
| Symbolic Reasoning | 2 | 0 | 2 |
| All Sources | 74 | 12 | 86 |

**2) Tasks should avoid excessive complexity within a single iteration.** Each iteration should be manageable for the LLM, as our goal is to isolate errors caused by length generalization failures rather than by inherent task complexity. Therefore, we exclude tasks with complex inputs (e.g., graphs, multi-dimensional data) or advanced math operations, which remain challenging for current LLMs.

Following these principles, we construct a dataset covering diverse domains, including code execution, number processing, logical and symbolic reasoning. Our data sources are as follows:

- **LeetCode Problems.**[1] Since most problems on LeetCode can be scaled with varying input sizes—primarily in terms of length—many of them are naturally suited for evaluating length generalization. For instance, in the task *LC Add Digits* ("repeatedly sum all digits in an integer until the result is a single digit"), we use the number of digits in the input to denote the inherent "length" of the task. Based on this criterion, we selected 74 tasks from the LeetCode platform.
- **NUPA.** NUPA is a benchmark designed to assess the basic number processing capabilities of LLMs [52]. While many tasks in NUPA are still overly challenging for current LLMs, we select four practical tasks feasible within the context length in terms of RFFT.
- **Big-Bench Hard.** The benchmark includes reasoning tasks considered challenging for LLMs [42]. We select 6 tasks that are suitable for length generalization evaluation.
- **Symbolic Reasoning.** We select "coin flip" and "last letter concatenation" from Wei et al. [48].

For data annotation, we engaged *skilled human annotators*, undergraduates majoring in Computer Science from top-tier universities in the nation, to write Python scripts that generate input-output traces for each task as shown in Figure 1. More detailed traces are shown in Appendix G.1. Each task is implemented as *a Python class to automatically generate samples of any given length*. All scripts underwent rigorous code reviews and filtering.

As shown in Table 1, our dataset includes 86 tasks in total, with examples of each domain presented in Appendix C.2, and the detailed description and length definition of each task in Appendix H.

## 4 Main Results

### 4.1 Experimental Setup

As introduced in Section 3.1, our Meta-RFFT involves *RF-pretraining* and *downstream adaptation*.

As shown in Table 1, in the *RF-pretraining* stage, we fine-tune the model on 74 tasks, aiming to develop a model that can strictly follow rules across multiple tasks and potentially transfer this capability to new tasks. For each task, 300 rule-following samples are generated for each length from 1 to 15, resulting in approximately 310k samples in total. We experiment on models of two different sizes: Qwen2.5-7B-Instruct and Qwen2.5-32B-Instruct [39]. The 7B model is fine-tuned with full-parameter training, while the 32B model is fine-tuned with PiSSA [29].

---

[1] https://leetcode.com/problemset

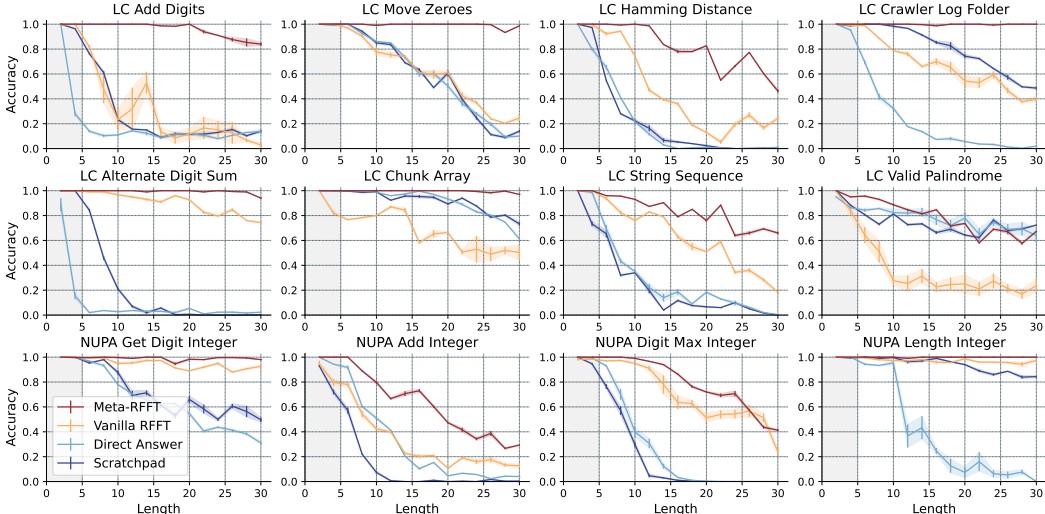

Figure 3: Length generalization performance of *direct answer*, *scratchpad*, *vanilla RFFT* and *Meta-RFFT* on LeetCode and NUPA tasks. The shaded region represents the in-distribution test results (length $\leq 5$), while the unshaded background corresponds to out-of-distribution lengths (length $\geq 6$). Here the base model is Qwen2.5-7B-Instruct.

In the *downstream adaptation* stage, we evaluate the models on 4 NUPA tasks and 8 LeetCode tasks of appropriate difficulty and practical significance respectively. The description of each downstream task is provided in Appendix C.3. We first train the models on data of lengths from 1 to 5 and then test their performance on out-of-distribution (OOD) lengths from 6 to 30 to evaluate the length generalization performance. For each task, we generate 1,000 samples for each length from 1 to 5, resulting in a total of 5k training samples. Both the 7B and 32B models are fine-tuned through PiSSA in the downstream adaptation stage. We evaluate models on 100 samples per length per task. Detailed training hyperparameters are provided in Appendix D.

**Baselines**  We use three baseline fine-tuning methods for comparison: *direct answer*, *scratchpad*, and *vanilla RFFT*. The input-output sequences are shown in Figure 1. The base model is fine-tuned directly on the target task on lengths from 1 to 5. To ensure fairness, all baseline methods use the identical downstream adaptation settings of Meta-RFFT, including training samples and steps. For long-CoT models, we evaluate using the same input prompts as the direct answer and scratchpad baselines in Figure 1, with `temperature=0` and `max_token=24,000`.

### 4.2 Meta-RFFT Enhances Task-Transferable Length Generalization

The length generalization performance of Qwen-2.5-7B-Instruct trained with direct answer, scratchpad, vanilla RFFT and Meta-RFFT is shown in Figure 3; the results of Qwen-2.5-32B-Instruct are shown in Figure 9 in Appendix E.1. Besides, we provide two unified metrics to give an overall performance comparison across all tasks: *ACC_Len30*, which measures the average accuracy at length 30 across tasks; and *Max_Len_90%*, which represents the maximum length (averaged across tasks) where the model maintains $\geq 90\%$ accuracy. The overall performance is summarized in Table 2.

Table 2: Overall metrics of performance of different methods across all 12 test tasks. Here, ACC_Len30 measures average accuracy at length 30; Max_Len_90% represents maximum length sustaining $\geq 90\%$ accuracy averaged across tasks.

|  | ACC_Len30 ($\uparrow$) | | Max_Len_90% ($\uparrow$) | |
|---|---|---|---|---|
| DeepSeek-R1-671B | 0.84 | | 19.17 | |
| QwQ-32B | 0.79 | | 19.33 | |
| ***Fine-Tuned Models*** | **7B model** | **32B model** | **7B model** | **32B model** |
| Direct Answer | 0.16 | 0.30 | 6.00 | 12.67 |
| Scratchpad | 0.30 | 0.41 | 7.50 | 11.17 |
| Vanilla RFFT | 0.40 | 0.67 | 9.33 | 18.17 |
| Meta-RFFT | **0.77** | **0.98** | **21.17** | **30.00** |

Overall, Meta-RFFT consistently outperforms other methods in length generalization for both 7B and 32B models. The 7B Meta-RFFT model exhibits a slower performance decline as sequence length increases, whereas direct answer, scratchpad, and vanilla RFFT suffer sharper drops when extrap-

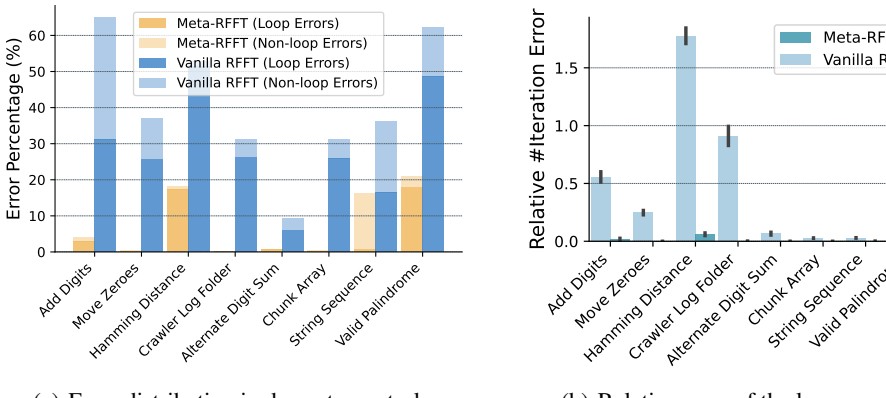

(a) Error distribution in downstream tasks.    (b) Relative error of the loop count.

Figure 4: Error analysis of vanilla RFFT and Meta-RFFT models.

olating to longer sequences. Notably, the 32B model with Meta-RFFT *achieves a Max_Len_90% of 30.00*, as shown in Table 2, demonstrating stable performance up to $6\times$ the training length. This suggests that the advantages of Meta-RFFT can be further developed with stronger base models.

Against cutting-edge long-CoT models, the 32B Meta-RFFT model improves ACC_Len30 by 14% over DeepSeek-R1-671B (84%) and 19% over QwQ-32B (79%). In Max_Len_90%, it surpasses both by over 10 lengths, underscoring superior robustness in length generalization. To provide insights for why cutting-edge long-CoT models fail in such seemingly intuitive tasks, such as long-integer addition, we provide an example error case of DeepSeek-R1-671B in Table 13 in Appendix F. More detailed results of long-CoT models are shown in Figure 11 in Appendix E.2.

**Error analysis: how does Meta-RFFT help length generalization?**    We analyze the errors of both vanilla RFFT and Meta-RFFT models and identify a primary cause of failure in length generalization: incorrect loop maintenance. Specifically, models often either terminate loops too early or fail to exit them, leading to repeated outputs until the context window limit is reached. As shown in Figure 4(a), errors due to incorrect loop counts form a substantial portion of total errors.

We hypothesize that RF-pretraining exposes the model to numerous rule-following examples involving loop control, enabling it to learn this sub-skill effectively. To test this, we measure the relative error between predicted and true iteration counts (Figure 4(b)). Meta-RFFT models exhibit significantly lower loop count errors than vanilla RFFT across tasks, confirming our hypothesis.

This reduction demonstrates that RF-pretraining improves the model's ability to manage iterative reasoning, which directly contributes to enhanced length generalization. Overall, the results indicate that length generalization transferability arises from mastering transferable computational patterns rather than task-specific ones.

**Analysis of Meta-RFFT's performance advantage over vanilla RFFT**    To understand why Meta-RFFT outperforms vanilla RFFT, we analyze their performance and training dynamics. Training curves during the downstream adaptation stage for both 7B and 32B models are shown in Figures 13 and 14 (Appendix E.4). Meta-RFFT models start with lower initial training loss, as their first-stage pretraining familiarizes them with the rule-following paradigm, which aids faster adaptation. Notably, while both methods *eventually reach similar training loss levels in most tasks*, Meta-RFFT consistently achieves better length generalization. This suggests that vanilla RFFT's limitations are not due to insufficient training. We further validate this by evaluating an intermediate Meta-RFFT checkpoint (Meta-RFFT-ckpt60), which has higher training loss than the converged vanilla RFFT. Even so, Meta-RFFT-ckpt60 outperforms vanilla RFFT (Figures 13, 17), confirming that Meta-RFFT's advantage arises from improved systematic generalization, rather than simply better fitting to training data.

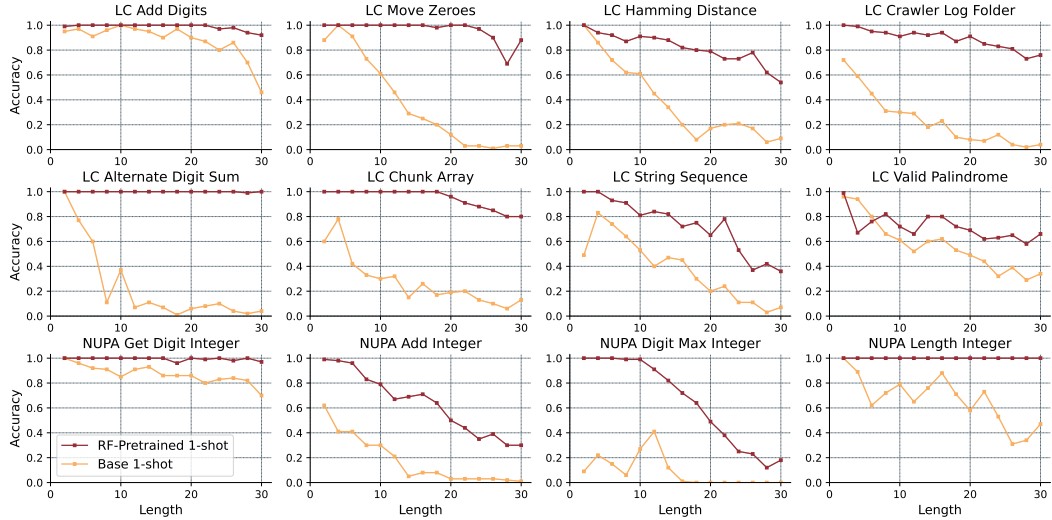

Figure 5: The 1-shot performance of the base model (Qwen-2.5-7B-Instruct), and the RF-pretrained model (RF-pretrained Qwen-2.5-7B-Instruct).

## 4.3 In-Context Learning

To enhance Meta-RFFT to be more user-friendly, we explore ICL settings in the downstream adaptation stage, as shown in Figure 2.

**Experimental Settings**  To enable the model to adapt to the in-context learning (ICL) paradigm, where few-shot exemplars are provided within the input, we include a 1-shot exemplar in each training sample. In-context learning requires the model to establish stronger correspondences between rules and execution traces, as it must learn to robustly follow a new rule from just a single exemplar. To improve this, we augment the training data with *synthetic tasks*, each assigned a unique rule. This approach increases task diversity and encourages the model to rely on the provided rules during training. Specifically, we manually design 22 code snippets and their corresponding rule-following outputs (details in Appendix G.2). For each sample, rules are dynamically composed by randomly selecting snippets, which enables arbitrary task generation with varied rules and outputs. Using this method, we create 100k synthetic samples and combine them with 310k samples from 74 tasks in Figure 2 to form the final ICL training dataset in the RF-pretraining stage.

**Results**  The ICL performance of 7B models is shown in Figure 5, with results of 32B models in Figure 10 in Appendix E.1. For both model sizes, the RF-pretrained model significantly outperforms the base model on downstream tasks in the 1-shot setting. Notably, the 32B model achieves a Max_Len_90% of 28.5, an improvement of 14.5 over the base model and even 10.3 over the vanilla RFFT model which is *fine-tuned on downstream tasks*. Crucially, this means an RF-pretrained model can achieve robust length generalization on *unseen* tasks with *only one exemplar*, which is a particularly valuable property for real-world deployment where task-specific fine-tuning is too costly.

## 5 Analysis

**What about following natural language rules?**  We use rules represented by Python programs in the previous sections due to their clarity, conciseness, and low ambiguity. However, rules can be expressed in various forms, and in everyday life, natural language is another primary medium for representing rules. We further investigate whether models trained on code-based rules during RF-pretraining can adapt to downstream tasks involving natural language-based rules. More specifically, to investigate whether the superior performance of Meta-RFFT on target tasks truly stems from a genuine understanding of general rules rather than overfitting to specific code statements like pop() and insert() during the RF-pretraining stage, we evaluate its adaptation to natural language rules in downstream tasks. Crucially, while Meta-RFFT is pretrained on code-based rules, we use natural language rules for fine-tuning in the downstream adaptation stage, ensuring no overlap in specific statements. An example of natural language rule is provided in Appendix G.3.

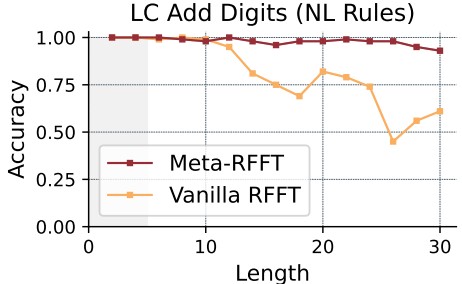
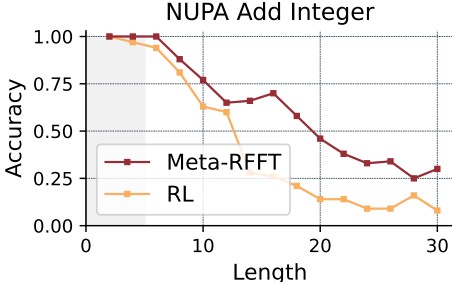

Figure 6: Vanilla RFFT and Meta-RFFT results on LC Add Digits. Here the model is Qwen2.5-7B-Instruct.

Figure 7: The performance of the Qwen2.5-7B after Meta-RFFT and RL on integer addition.

As shown in Figure 6, Meta-RFFT still significantly outperforms vanilla RFFT on *LC Add Digits* across lengths 12-30. This confirms that Meta-RFFT's advantage arises not from memorizing code syntax but from acquiring a meta rule-following capability that enhances length generalization.

**What about reinforcement learning for rule following?** While reinforcement learning (RL) has shown success in general reasoning, we argue that SFT is better suited for enforcing strict rule-following behavior. Unlike RL with outcome reward, which only optimizes for *final-answer correctness*, SFT directly *maximizes the likelihood of the model generating rule-compliant intermediate reasoning steps*, which is crucial for strict rule adherence. To validate this, we optimize the base model using Proximal Policy Optimization (PPO) [40]. We provide training details in Appendix D.3. The RL variant achieves lower accuracy, confirming that pure outcome supervision fails to ensure rule compliance, as shown in Figure 7. Instead of learning rule-following behavior, the model tends to shortcut to direct answers, impairing generalization to longer reasoning chains (see Table 14 in Appendix F for example error traces). Moreover, while RL training requires 64 H800 GPU hours per task, our Meta-RFFT method achieves better performance in just 2.4-4.4 GPU hours, demonstrating significantly higher efficiency. The detailed training compute are listed in Table 8 in Appendix D.2.

**Comparison to instruction following** Previous studies have focused on enhancing the instruction-following capabilities of LLMs [22, 43, 50, 47], a fundamental ability for their practical application. However, existing instruction-following models still struggle to adhere to complex rules. In the length generalization scenarios,

Table 3: Overall comparison between *Meta-RFFT* and *Tulu3*. The base model is Llama3.1-8B.

|  | ACC_Len30 (↑) | Max_Len_90% (↑) |
|---|---|---|
| Tulu3 | 0.16 | 0.67 |
| Meta-RFFT | **0.38** | **11.67** |

as the inherent "length" of a problem increases, the corresponding rules grow more complex, and current instruction-tuned models often fail to strictly follow these rules strictly. We compare current instruction-following methods and Meta-RFFT from two perspectives.

First, in our previous experiments, including fine-tuning (7B: Figure 3, 32B: Figure 9) and ICL (7B: Figure 5, 32B: Figure 10), our baselines are advanced instruction-tuned models (Qwen2.5-7B-Instruct and Qwen2.5-32B-Instruct). However, they exhibit poor length generalization performance, especially when provided with 1-shot rule-following exemplar, they fail to strictly adhere to the given rules. In contrast, after Meta-RFFT, these models demonstrate significant improvements.

Second, we compare Meta-RFFT and instruction tuning with Llama-3.1-8B [16] as the base model. For instruction tuning, we evaluate Llama-3.1-Tulu-3.1-8B [22], a version fine-tuned on the Tulu3 dataset to enhance instruction following, with 1-shot prompting to provide a rule-following exemplar. As Table 3 shows, the instruction-tuned model fails to consistently follow rules during reasoning, while Meta-RFFT achieves significantly better performance. Full results are in Figure 12 (Appendix E.3).

**Generalization to practical tasks** To validate the practical value of Meta-RFFT beyond symbolic tasks, we evaluate our method on a real-world scenario: the Airline Baggage Cost Estimation task from Zhou et al. [59]. This task requires models to interpret and apply natural language regulations—a capability that cannot be fully reduced to formal code execution.

We train models on samples with length 1–8, and report test accuracy on both in-distribution lengths (2–8) and OOD lengths (10–16). As shown in Figure 8, Meta-RFFT significantly outperforms vanilla RFFT in the practical domain, while current LLMs perform poorly on it, and even strong models like Claude-3.5 and o1-preview degrade rapidly with task length growing.

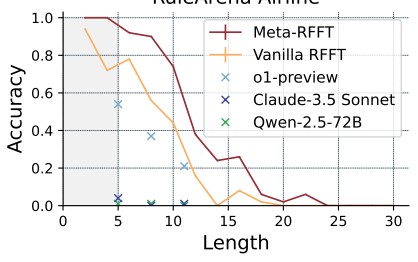

Figure 8: Length generalization performance on the airline task.

**Impact of Meta-RFFT on general model capabilities** The extensive RF-pretraining involved in Meta-RFFT introduces a potential risk of distribution shift, which can lead to catastrophic forgetting [14, 27] of the model's general capabilities. However, this issue can typically be mitigated by mixing some pretraining corpus during fine-tuning.

To preserve the model's general capabilities, we augmented the meta-rule-following fine-tuning data with training samples from the GSM8K mathematical reasoning dataset [7]. Specifically, for each training example in GSM8K, we use the base model Qwen2.5-7B-Instruct to generate 10 candidate answers and filter the correct responses as the training set, forming a filtered dataset of 7182 samples.

The results on length generalization (averaged across 8 LeetCode tasks) are shown in Table 4, and the performance on GSM8K test set are shown in Table 5. We observe that merely mixing 7k samples (compared to 310k Meta-RFFT data) effectively recovers the model's reasoning performance on GSM8K, while maintaining the strong multi-task length generalization ability. Notably, Meta-RFFT still significantly outperforms vanilla RFFT, demonstrating that general capabilities can be preserved with minimal additional data.

We extend this analysis to other domains, including common-sense reasoning (ARC [6]) and instruction-following alignment (Alpaca-Eval [25]). The results, presented in Appendix E.8 (Tables 12 and 11), consistently show that all evaluated general capabilities can be recovered with minimal fine-tuning data while the model retains robust length generalization.

Table 4: Length generalization performance of Meta-RFFT trained with mixed data.

|  | ACC_Len30 ($\uparrow$) | Max_Len_90% ($\uparrow$) |
|---|---|---|
| Vanilla RFFT | 0.34 | 7.00 |
| Meta-RFFT | 0.82 | 22.75 |
| w/ GSM8K | 0.79 | 21.25 |

Table 5: Performance on GSM8K test set of Meta-RFFT trained with mixed data.

|  | GSM8K ACC |
|---|---|
| Base Model | 87.95 |
| Meta-RFFT | 87.80 |

# 6 Conclusion

We make an initial attempt to enhance *cross-task* length generalization in the post-training stage of LLMs. To this end, we construct a large-scale length generalization dataset comprising 86 diverse tasks spanning diverse domains, which significantly expands length generalization research beyond traditional tasks like addition and sorting. Our experiments show that Meta-RFFT on 74 of these tasks facilitates strong length generalization on unseen tasks with minimal downstream fine-tuning or 1-shot prompting, even surpassing advanced long-CoT models. Through extensive analysis, we show that this transferability arises from the model's ability to learn shared computational primitives rather than relying on task-specific patterns. Additionally, we verify that models pretrained on code-based rules can successfully adapt to natural language rules in downstream tasks.

# Acknowledgment

This work is supported by the National Key R&D Program of China (2022ZD0160300), Kunpeng&Ascend Center of Excellence, Peking University, and CCF-Tencent Rhino-Bird Open Research Fund.

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

# List of appendices

# Appendices

## A  Limitations

In this work, we construct a large-scale length generalization dataset spanning diverse tasks and propose Meta-RFFT to enhance cross-task length generalization in LLMs. While this serves as an initial step toward understanding multi-task length generalization in LLMs, our current study has several limitations. First, to isolate errors attributable to length generalization failures (rather than inherent task complexity), we restrict our experiments to controlled settings involving code execution, numerical processing, and symbolic / logical reasoning tasks. Consequently, our framework does not yet address more complex, real-world long-horizon reasoning domains, such as legal judgment generation or multi-step workflow execution. Furthermore, defining "length" as a metric for problem complexity in such practical scenarios remains an open challenge. Extending length generalization to these domains, where models trained on simpler instances must generalize to harder problems, presents a promising direction for future research.

## B  Impact Statements

Our work focuses on establishing a relationship between rules and their corresponding instances within LLMs. We aim to enhance model performance on downstream tasks by training the base model on a wide range of rule-following tasks. Current training strategies fall short in enabling models to fully grasp the rules that humans have summarized or proposed that exist in the pre-training corpus. As a result, while LLMs can easily recall rules, they often struggle to apply these rules strictly to specific instances. Our proposed Meta-RFFT takes an initial step towards strengthening models into meta rule followers, a development that is crucial for improving both the reasoning capabilities and learning efficiency of these models.

Teaching LLMs to follow rules also aligns with societal demands. By ensuring that LLMs can reliably adhere to rules, we contribute to the development of AI systems that are more aligned with human values, ethical standards, and practical applications, ultimately fostering trust and safety in their deployment.

## C  Dataset Overview

### C.1  Data Annotations

We engaged skilled human annotators (undergraduate students majoring in Computer Science from top-tier universities) to write Python scripts generating input-output traces for each task. Each task was implemented as a Python class to enable automated sample generation at specified lengths.

Prior to annotation, we conducted a **detailed training session using 12 exemplar tasks** to ensure consistency and quality. Annotators received step-by-step tutoring and were required to pass a qualifying test on a sample task before proceeding.

Following annotation, all scripts underwent rigorous validation, including manual code review and automated filtering, to eliminate errors. All annotators were compensated fairly for their work.

### C.2  Rule-Following Input Examples

We present a question example for each reasoning domain in Table 6.

### C.3  Downstream Tasks Description

The descriptions of 12 selected downstream tasks are listed as follows:

- **LC Add Digits:** Given an integer, repeatedly sum its digits until the result is a single digit.
- **LC Move Zeroes:** Given a list of integers, move all zeros to the end while preserving the relative order of the non-zero elements.

Table 6: Input example in rule-following format for each category.

| LeetCode | NUPA |
|---|---|
| Follow the given rule to solve the question.
rule:

```python
def moveZeros(nums):
    num_zero = 0
    result = []
    while nums:
        number = nums.pop(0)
        if number != 0:
            result.append(number)
        else:
            num_zero += 1
    i = 0
    while i < num_zero:
        result.append(0)
        i += 1
    return result
```

Q: Given an integer array [0, 16], move all zeros to the end while preserving the relative order of the non-zero elements. | Follow the given rule to solve the question.
rule:

```python
def add(num1, num2):
    result = ''
    carry = 0
    # Main Loop
    while num1 or num2:
        digit1 = int(num1[-1]) if num1
            ↪   else 0
        digit2 = int(num2[-1]) if num2
            ↪   else 0
        total = digit1 + digit2 +
            ↪ carry
        result = str(total%10) +
            ↪ result
        carry = total//10
        num1 = num1[:-1] if num1 else
            ↪ num1
        num2 = num2[:-1] if num2 else
            ↪ num2
    if carry:
        result = str(carry) + result
    result = result.lstrip('0') or '0'
    return result
```

Q: Add two numbers: 123 + 4567. |
| **Big-Bench Hard** | **Symbolic Reasoning** |
| Follow the given rule to solve the question.
rule:

```python
def navigate(moves):
    # Initialize Location
    loc = [0, 0]
    # Main Loop
    while moves:
        move = moves.pop(0)
        if move[0] == "left":
            loc[0] -= move[1]
        elif move[0] == "right":
            loc[0] += move[1]
        elif move[0] == "forward":
            loc[1] += move[1]
        elif move[0] == "backward":
            loc[1] -= move[1]
    return loc == [0, 0]
```

Q: If you follow these instructions, do you return to the starting point? Always face forward. Take 2 steps left. Take 2 steps forward. In short, the moves are as follows: [('left', 2), ('forward', 2)]. | Follow the given rule to solve the question.
rule:

```python
def coin_flip(flips):
    # Initialize Coin State
    heads_up = True
    # Main Loop
    while flips:
        flip = flips.pop(0)
        if flip:
            heads_up = not heads_up
        else:
            pass
    return heads_up
```

Q: A coin is heads up. Carrillo does not flip the coin. Cunningham flips the coin. Is the coin still heads up? In short, the situation of 2 people flipping coins is as follows: [False, True]. |

- **LC Hamming Distance:** The Hamming distance between two integers is the number of positions at which the corresponding bits are different. Given two integers in binary representation, return their Hamming distance.
- **LC Crawler Log Folder:** Determine the final folder after performing the operations in the given list, where `../` moves up one level, `./` stays in the current folder, and `x/` moves into folder `x`.
- **LC Alternate Digit Sum:** Given a positive integer where the most significant digit has a positive sign, and each subsequent digit has the opposite sign of its adjacent digit, return the sum of these signed digits.
- **LC Chunk Array:** Given array and chunk size, split the array into subarrays of a given size.
- **LC String Sequence:** Given a target string, return a list of all strings that appear on the screen in order, using the minimum key presses. Key 1 appends "a" to the string, and Key 2 changes the last character to its next letter in the alphabet.
- **LC Valid Palindrome:** Given a string s, return true if it is a palindrome after removing all non-alphanumeric characters and converting it to lowercase; otherwise, return false.
- **NUPA Get Digit Integer:** Get the digit at the given position (from left to right, starting from 0).
- **NUPA Add Integer:** Add two integers.
- **NUPA Digit Max Integer:** Compare two numbers digit by digit and return the larger digit at each position, treating any missing digits as 0.
- **NUPA Length Integer:** Find total number of digits of the given integer.

## D  Training Details

### D.1  Training Hyperparameters

Table 7 shows the training parameters for the RF-pretraining stage and the adaptation stage of the Qwen-7B and Qwen-32B models. In the RF-pretraining stage, we use data samples with a length of 31 from the training set as the validation set and the early stop strategy is applied based on the validation loss, which results in different training steps. Considering early stopping, the number of training data samples for the 7B and 32B models in the RF-pretraining stage is 179k and 205k, respectively.

Since the RF-pretraining stage involves fine-tuning across numerous tasks, we train more model parameters during this stage. The 7B model uses full parameter fine-tuning, while due to computational resource constraints, the 32B model employs PiSSA with a large rank of 32. In the adaptation stage, where fine-tuning is performed on different target tasks, we use PiSSA with a relatively small rank 8.

Table 7: The training hyperparameters for the RF-pretraining stage and the adaptation stage.

| Hyperparameters | RF-Pretrain | | Adaptation | |
|---|---|---|---|---|
| | Qwen-7B | Qwen-32B | Qwen-7B | Qwen-32B |
| Training Steps | 800 | 700 | 156 | 156 |
| Num of Epoch | 1 | | | |
| Learning Rate | 1e-5 | | | |
| Batch Size | 256 | | 32 | |
| Fine-Tuning Method | full fine-tune | | PiSSA | |
| PiSSA Rank | / | 32 | 8 | |

### D.2  Compute Resources

We list the training compute of RF-pretraining and downstream fine-tuning in Table 8. We conduct all the experiments on NVIDIA H800 Tensor Core GPUs.

While Meta-RFFT does require initial pretraining, our analysis shows it becomes significantly more efficient than vanilla RFFT when deployed across multiple tasks.

- **Computation Efficiency:** For the 7B model, the pretraining cost becomes justified after just 42 downstream tasks—a threshold quickly exceeded in practice. For 32B model, the number is 72.
- **Data Efficiency**: Meta-RFFT eliminates the need for task-specific fine-tuning data. Meta-RFFT works in the context of in-context learning with just the rule and one demonstration.

Table 8: Training compute of RF-pretraining and downstream fine-tuning of both 7B and 32B models.

| Model | Training Stage | Training hours | GPU Num | GPU Memory | GPU hours |
|-------|----------------|----------------|---------|------------|-----------|
| 7B | RF-pretrain | 18 | 8 | 80G | 144.0 |
| 7B | downstream | 0.6~1.1 | 4 | 80G | 2.4~4.4 |
| 32B | RF-pretrain | 22.3 | 32 | 80G | 713.6 |
| 32B | downstream | 2.0~3.0 | 4 | 80G | 8.0~12.0 |

### D.3 Reinforcement Learning Settings

We utilize Proximal Policy Optimization (PPO) without KL-regularization as our reinforcement learning algorithm. PPO updates the policy parameters $\theta$ to maximize the expected cumulative reward, while simultaneously updating the value function parameters $\phi$ by minimizing the value loss. This is achieved by optimizing the following objective functions:

$$\mathcal{J}_{\text{PPO}}(\theta) = \mathbb{E}_{t,s_t,a_t \sim \pi_{\theta_{\text{old}}}} \left[ \min \left( \frac{\pi_\theta(a_t|s_t)}{\pi_{\theta_{\text{old}}}(a_t|s_t)} \hat{A}_t, \text{clip} \left( \frac{\pi_\theta(a_t|s_t)}{\pi_{\theta_{\text{old}}}(a_t|s_t)}, 1 - \epsilon, 1 + \epsilon \right) \hat{A}_t \right) \right], \quad (1)$$

$$\mathcal{J}_{\text{value}}(\phi) = \frac{1}{2} \mathbb{E}_{t,s_t,a_t \sim \pi_{\theta_{\text{old}}}} \left[ (V_\phi(s_t) - R_t)^2 \right], \quad (2)$$

We provide an outcome reward, where the model receives a reward of 1 only when its output answer is completely correct, and 0 otherwise. Key hyperparameters used in our implementation are listed in Table 9.

Table 9: Training Hyperparameters for PPO.

| Hyperparameter | Value |
|----------------|-------|
| Actor Learning Rate | 1e-6 |
| Critic Learning Rate | 9e-6 |
| Train Batch Size | 1024 |
| Rollout Batch Size | 256 |
| GAE parameter $\lambda$ | 1.0 |
| Discount Factor $\gamma$ | 1.0 |
| Clipping Parameter $\epsilon$ | 0.2 |
| KL Coefficient | 0.0 |

## E Additional Results

### E.1 Results of 32B Models

**Fine-tuning results.** In Figure 9, we list the length generalization performance of Qwen-2.5-32B-Instruct fine-tuned through the following methods: *direct answer*, *scratchpad*, *vanilla RFFT* and *Meta-RFFT*. *Meta-RFFT* significantly outperforms the rest of the methods, showing that rule-following is a meta ability that can be mastered by large-scale RF-pretrain and can benefit length generalization greatly.

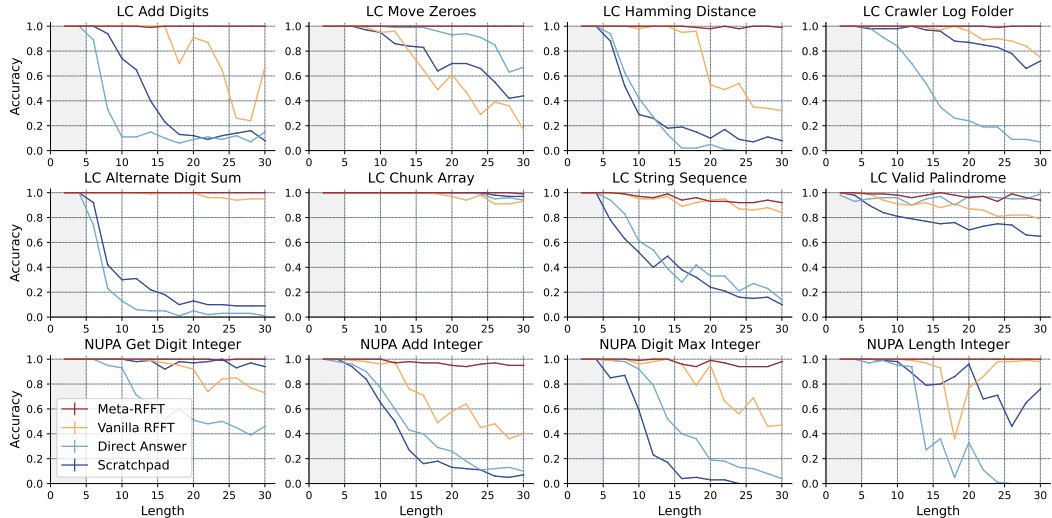

Figure 9: Length generalization performance of *direct answer*, *scratchpad*, *vanilla RFFT* and *Meta-RFFT* on LeetCode and NUPA tasks. Here the experiments are all conducted on Qwen-2.5-32B-Instruct.

**In-context learning results.** In Figure 10, we show the in-context learning performance of both the base model and the RF-pretrained model. Here, the base model we use is Qwen-2.5-32B-Instruct. The RF-pretrained model outperforms the base model by a large margin in the context of 1-shot learning on downstream tasks. RF-pretraining shows a positive transfer to the in-context rule-following capabilities in downstream tasks.

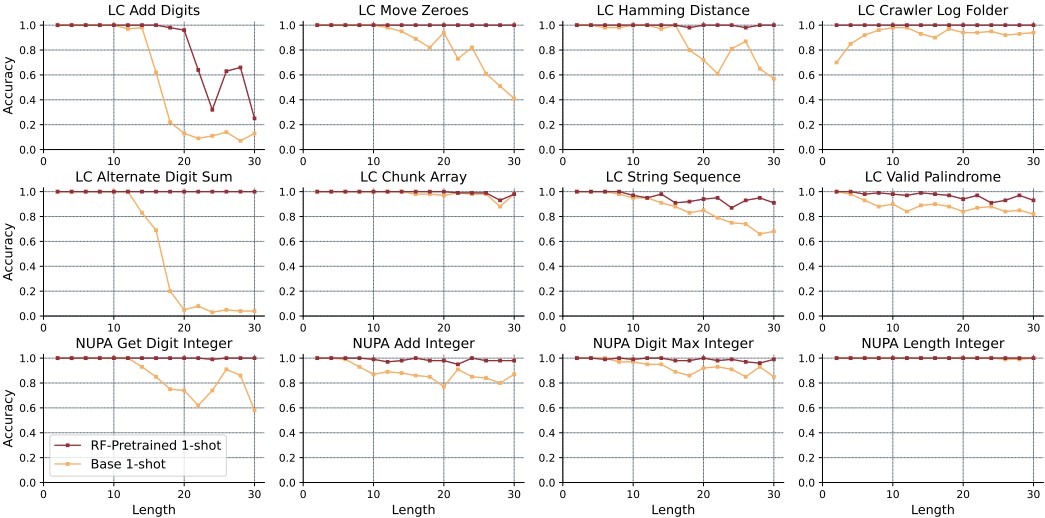

Figure 10: The figure shows the 1-shot performance of the base model (Qwen-2.5-32B-Instruct), and the RF-pretrained model (RF-pretrained Qwen-2.5-32B-Instruct).

## E.2 Length Generalization of Long-CoT Models

In Figure 11, we show the length generalization performance of long-CoT models, including DeepSeek-R1-671B, DeepSeek-R1-Distill-7B and QwQ-32B. Meta-RFFT-enhanced Qwen-32B shows superior performance regarding length generalization with comparable or even much smaller parameter size.

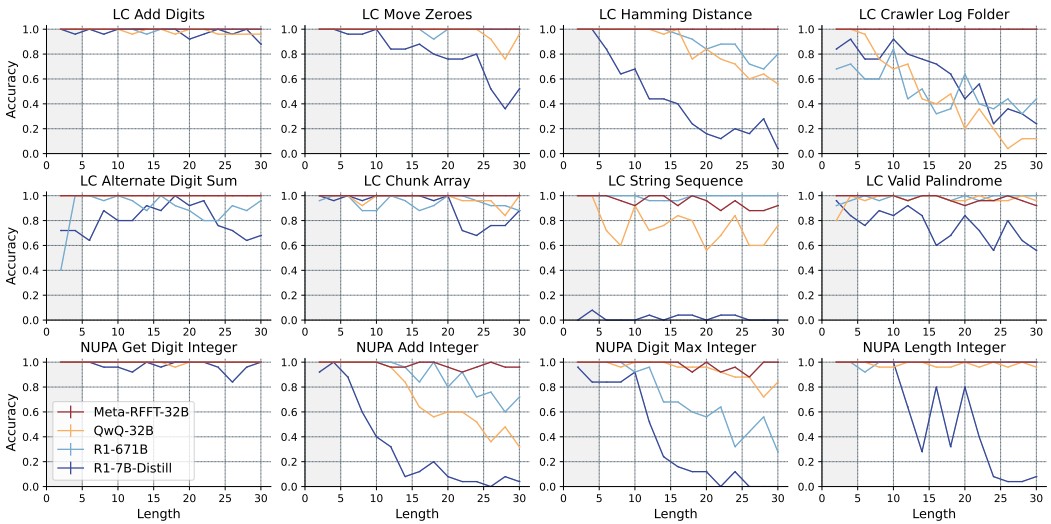

Figure 11: Length generalization performance of long-CoT models, including DeepSeek-R1-671B, DeepSeek-R1-Distill-7B and QwQ-32B.

### E.3 Comparison between Meta-RFFT and Instruction Following

We show the performance of Meta-RFFT-enhanced Llama-3.1-8B and the same model after instruction-tuning on Tulu3 in Figure 12. For Meta-RFFT, we fine-tune the model on samples of lengths from 1 to 5.

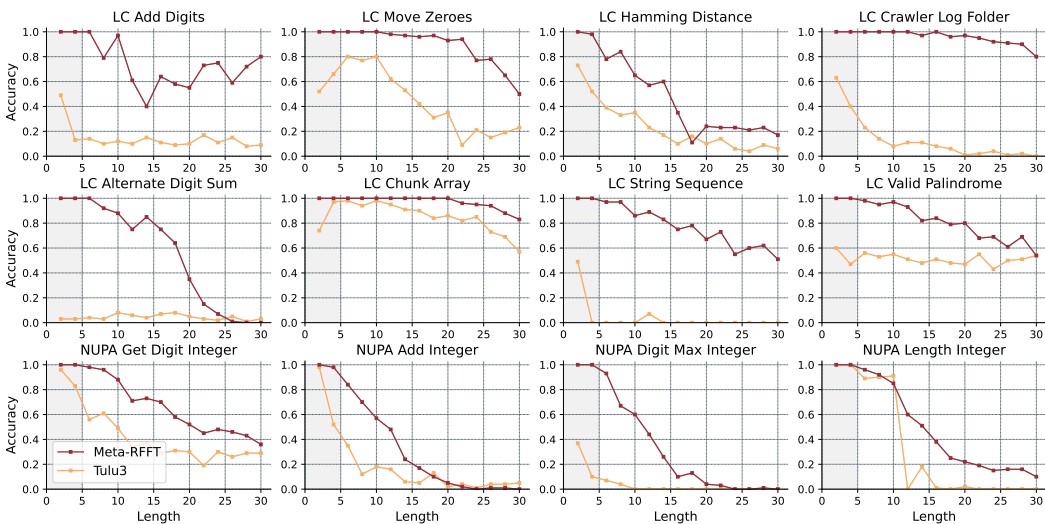

Figure 12: Comparison between *Meta-RFFT* and instruction following method *Tulu3*, using Llama-3.1-8B as the base model.

### E.4 Training Curves

We show the training loss curves of the downstream adaptation stage of the 7B base model and the 32B base model respectively in Figure 13, Figure 14. The figures show that models trained with Meta-RFFT exhibit lower initial training loss compared to vanilla RFFT, as the former is already familiar with the rule-following paradigm due to the first-stage pretraining. This allows Meta-RFFT models to fit the training samples more quickly during the adaptation stage. As training progresses, the training loss of vanilla RFFT and Meta-RFFT models converges to similar levels in most tasks. This indicates that the gap in length generalization performance between Meta-RFFT and vanilla

RFFT is not due to the latter's in- ability to fit the training data. More detailed discussions are put in Section 4.2.

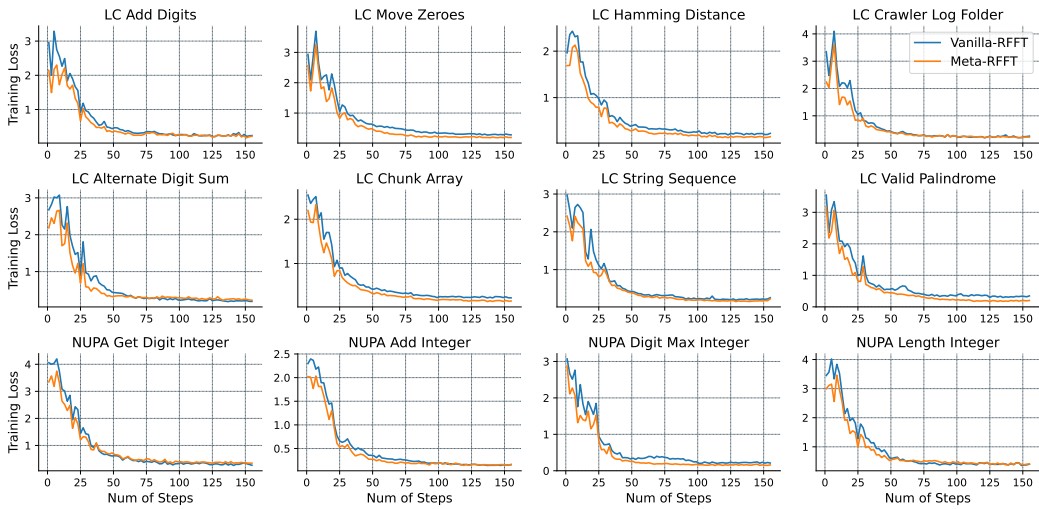

Figure 13: Training curves of Qwen2.5-7B-Instruct in the downstream adaptation stage on LeetCode tasks and NUPA tasks. The figure shows both the training curves of vanilla RFFT and the second training stage of Meta-RFFT.

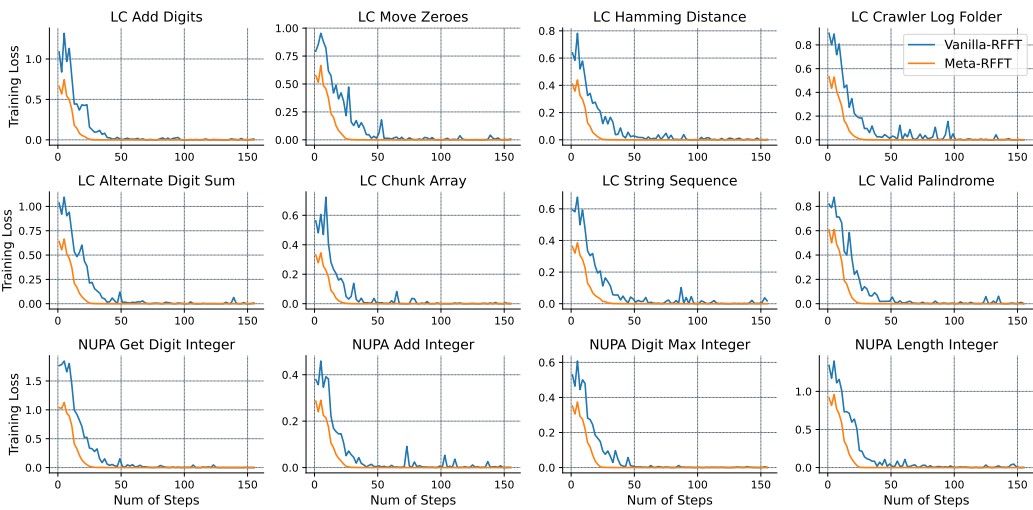

Figure 14: Training curves of Qwen2.5-32B-Instruct in the downstream adaptation stage on LeetCode tasks and NUPA tasks. The figure shows both the training curves of vanilla RFFT and the second training stage of Meta-RFFT.

Besides, we conduct repeated experiments in the stage of downstream fine-tuning. We show the training curves of different random seeds of 7B RF-pretrained models in the adaptation stage training in Figure 15, indicating that the adaptation stage training is stable across different seeds.

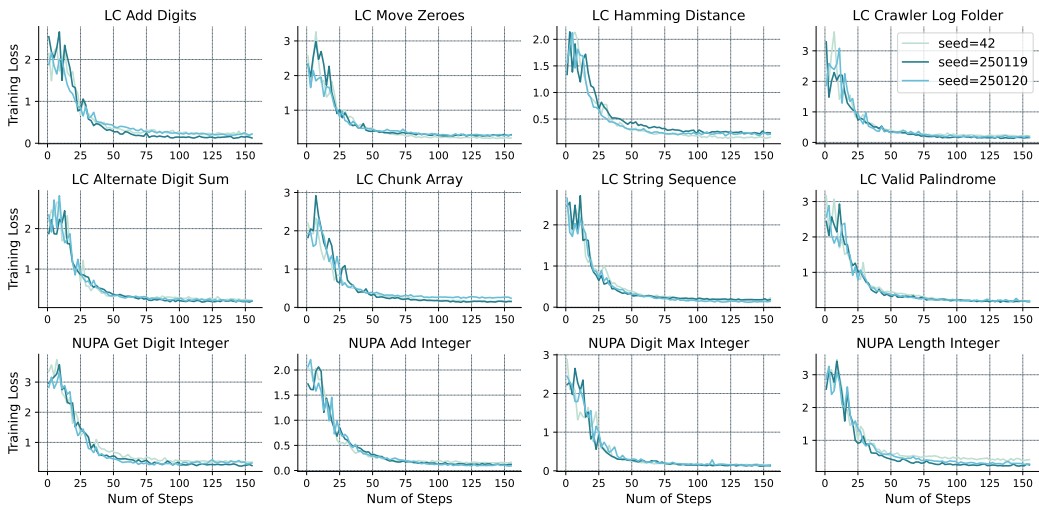

Figure 15: Training curves of the adaptation stage of Meta-RFFT using different random seeds. Here we show the results of Qwen2.5-7B-Instruct.

## E.5 Effects of Data Dize on RF-pretraining and Downstream Adaptation

**The effect of the data size in RF-pretraining.** We select several checkpoints from the RF-pretraining stage after the training loss has converged and perform downstream adaptation on these checkpoints. The performance of these checkpoints on the corresponding tasks after fine-tuning is presented in Figure 16. In the early stages, as training progresses, the model's final performance gradually improves. However, after reaching a certain number of steps, the model's performance stabilizes and no longer shows significant improvement. The models do not show signs of "grokking" during the RF-pretraining stage.

**The effect of the data size in downstream adaptation** For the downstream adaptation stage, we also analyze the effects of data size on performance. We select several checkpoints after the training loss has converged. The results of these checkpoints are presented in Figure 16. Similar to the RF-pretraining stage, in the early phases, the model's performance improves as the data size increases. However, after reaching a certain number of steps, the model's performance stabilizes and no longer shows significant improvement, with no evidence of grokking observed.

For RF-pretraining stage, we select several checkpoints after the training loss has converged and perform downstream adaptation on these checkpoints. The length generalization performance of these checkpoints on

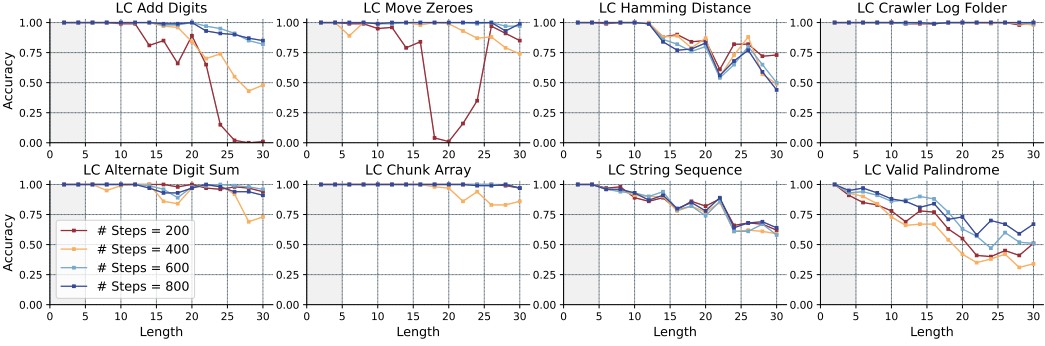

Figure 16: Effects of training steps in the RF-pretraining stage.

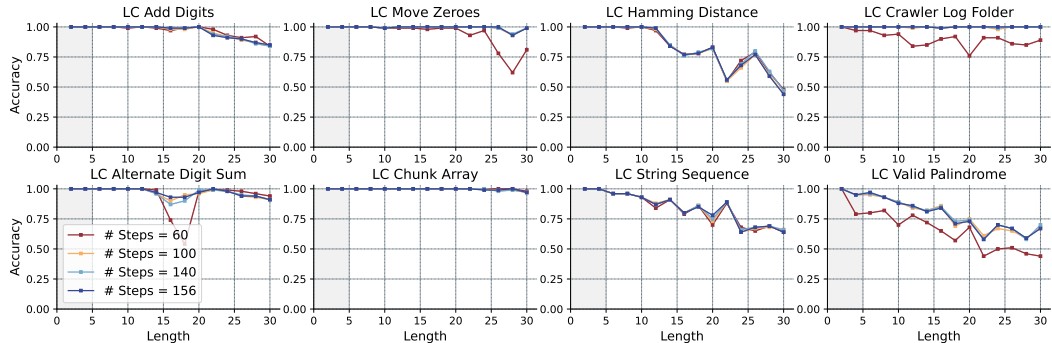

Figure 17: Effects of training steps in the downstream adaptation stage of Meta-RFFT.

## E.6 Effects of Number of Tasks in RF-Pretraining Stage

We show in Figure 18 that when fine-tuned with only 1 task in RF-pretraining stage, the model does not perform as well as Meta-RFFT-ed on 74 diverse tasks. This demonstrates that to enable robust multi-task length generalization, a large-scale length generalization dataset is necessary.

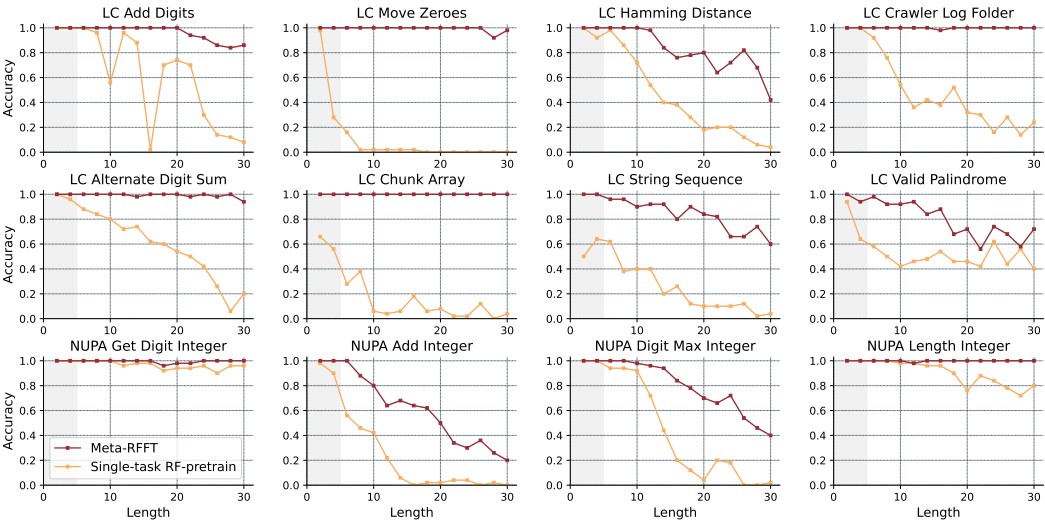

Figure 18: Performance when RF-pretrained on a single task versus 74 diverse tasks, showing the benefit of large-scale multi-task pretraining for length generalization.

## E.7 Effects of RF-pretraining Tasks on Downstream Performance

The RF-pretraining stage originally includes tasks from three datasets: LeetCode, Big-Bench Hard, and Symbolic Reasoning. To assess the impact of RF-pretraining tasks on downstream performance, we conduct an ablation where the model is RF-pretrained only on the LeetCode subset.

Here we evaluate on 4 downstream tasks, including 2 LeetCode tasks (Move Zeros & Valid Palindrome) and 2 NUPA tasks (Add Integer & Digit Max Integer), all of which are unseen during RF-pretraining.

The results are shown in Table 10. We observe a slight performance drop when excluding Big-Bench Hard and Symbolic Reasoning, likely due to reduced task diversity during pretraining.

## E.8 Effects of Meta-RFFT on General Model Capabilities

Similar to our experiments on GSM8K introduced in Section 5, we mix general data into the downstream fine-tuning alongside the original rule-following data to show that general ability

Table 10: Results of Meta-RFFT with LeetCode-only pretraining on multiple downstream benchmarks.

| Model | ACC_Len30 (%) | Max_Len_90% |
|---|---|---|
| **Results on LC and NUPA** | | |
| Meta-RFFT (complete pretrain dataset) | 56.0 | 16.0 |
| Meta-RFFT (LeetCode only) | 51.0 | 12.5 |
| Vanilla RFFT | 19.0 | 5.5 |
| **Results on SR and BBH** | | |
| Meta-RFFT (LeetCode only) | 68.0 | 20.7 |
| Vanilla RFFT | 37.0 | 9.3 |

is preserved. The results of the model's general ability are shown in Table 11, and the length generalization results are shown in Table 12.

Regarding length generalization results, we test on 2 downstream task: LC Valid Palindrome and NUPA Add Integer. Our findings indicate that incorporating 6,872 samples from ARC [6] and 8,000 samples from Alpaca-Eval [25] effectively restored the model's performance on these general benchmarks while maintaining robust multi-task length generalization.

On Alpaca-Eval, we observed a significant increase in the standard win rate. However, this improvement is partially attributable to the model producing longer outputs, which are favored by this metric. To provide a more balanced assessment, we also report the length-controlled win rate introduced by Dubois et al. [12].

Table 11: Evaluation of the general capabilities of Meta-RFFT with mixed training on ARC and Alpaca-Eval.

| Model | ARC | Alpaca-Eval | |
|---|---|---|---|
| | Acc | Win Rate | Length Controlled Win Rate |
| Base Model | 87.88 | 16.77 | 25.61 |
| Meta-RFFT | 88.92 | 23.54 | 23.90 |

Table 12: Length generalization performance of Meta-RFFT with mixed finetuning that incorporates general data.

| | ACC_Len30 ($\uparrow$) | Max_Len_90% ($\uparrow$) |
|---|---|---|
| Vanilla RFFT | 0.20 | 1.0 |
| Meta-RFFT | 0.58 | 8.0 |
| w/ ARC | 0.63 | 8.0 |
| w/ Alpaca | 0.60 | 8.0 |

# F Error Cases

## F.1 Error Case of DeepSeek-R1 on Addition

In Table 13, we present an error case where DeepSeek-R1 fails to correctly solve a 30-digit integer addition problem. In this case, R1 makes two mistakes: (1) **incorrect splitting:** R1 attempts to split the 30-digit integer into two parts, including a 17-digit segment and a 13-digit segment. However, the first segment actually contains 18 digits, and the two parts overlap by one digit. Despite this mistake in the reasoning trace, R1 miraculously corrects the error in the final answer; (2) **copying error**: R1 makes an copy error in the final answer, copying "7711" to "7771".

## F.2 Error Case of Models Trained with RL

While RL has shown success in general reasoning, we argue that SFT is better suited for enforcing strict rule-following behavior. Unlike RL with outcome reward, which only optimizes for final-answer correctness, SFT directly maximizes the likelihood of the model generating rule-compliant intermediate reasoning steps, which is crucial for strict adherence to structured rules.

We use RL (PPO) to optimize rule-following. The model does not learn rule-following behavior and instead exhibit tendencies to answer directly, thus fails to generalize to longer answers, as shown in Table 14.

Table 13: An error example of DeepSeek-R1 on long integer addition. We only show the summarization part as the thinking part is too long. R1 tricks itself during splitting the 30-digit number.

| Input | Output |
|---|---|
| Q: Add two numbers:
8898509165985 +
897653763194685878813426157006 | To add the numbers 8898509165985 and 897653763194685878813426157006, we need to align them properly and perform the addition step by step, considering carryovers.

1. Identify the digits of each number:
    - The first number is 13 digits:
        ↪ 8,898,509,165,985
    - The second number is 30 digits:
        ↪ 897,653,763,194,685,878,813,426,157,006

2. Split the second number into two parts:
    - The first 17 digits:
      897,653,763,194,685,878 (here R1 list 18 digits)
    - The last 13 digits: 8,813,426,157,006

3. Add the first number to the last 13 digits of
    ↪ the second number:
    - Last 13 digits of the second number:
        ↪ 8,813,426,157,006
    - First number: 8,898,509,165,985

4. Perform the addition of the last 13 digits:
    - Align and add the digits from right to left,
        ↪   considering carryovers:
      ```
        8,898,509,165,985
      + 8,813,426,157,006
      --------------------
       17,711,935,322,991
      ```
    - The result of adding these 13-digit numbers
        ↪ is 17,711,935,322,991, which is a 14-
        ↪ digit number. This results in a
        ↪ carryover of 1 to the higher digits.

5. Adjust the first part of the second number:
    - The first 17 digits of the second number:
        ↪ 897,653,763,194,685,878 (still 18 digits)
    - Add the carryover 1: 897,653,763,194,685,878
        ↪   + 1 = 897,653,763,194,685,879

6. Combine the adjusted first part with the
    ↪ result of the addition:
    - The adjusted first part:
        ↪ 897,653,763,194,685,879
    - The last 13 digits of the sum (after
        ↪ removing the carryover):
        ↪ 7,711,935,322,991

7. Concatenate the results:
    - The final result is 897,653,763,194,685,879
        ↪ followed by 7,711,935,322,991, which
        ↪ gives the 30-digit number.

The result of the addition is: 8976537631946858877771935322991


Correct answer should be: 897653763194685887711935322991 |

Table 14: An error example of Qwen2.5-7B-RL, which is Qwen2.5-7B trained by RL on rule-following samples.

| Input | Output |
|---|---|
| Follow the given rule to solve the question. rule: 

 ```python def add(num1, num2):     result = ''     carry = 0     # Main Loop     while num1 or num2:         digit1 = int(num1[-1]) if num1           ↪ else 0         digit2 = int(num2[-1]) if num2           ↪ else 0         total = digit1 + digit2 + carry         result = str(total%10) + result         carry = total//10         num1 = num1[:-1] if num1 else num1         num2 = num2[:-1] if num2 else num2     if carry:         result = str(carry) + result     result = result.lstrip('0') or '0'     return result ``` 

 Q: Add two numbers: 9592473675434172121187562571510 + 440884190197. | The given rule defines a function add(num1, num2) that adds two numbers represented as strings. I will use this function to add the two given numbers: 9592473675434172121187562571510 and 440884190197. 9592473675434172121187562615607 


 Correct answer should be: 9592473675434172121628446761707 |

## G   Dataset Construction

### G.1   Rule-following Dataset Annotation

To generate the rule-following output for each task, the rules necessary to accomplish the task first need to be provided in the input (examples are shown in Table 6), which can be generated by an LLM. Subsequently, a data generation script is manually written for each task to produce the intermediate execution steps of the rule for any given input. With the rule-following output generation script and the input data generator, an arbitrary number of data samples can be generated. Here is an example of one data sample in the rule-following format.

Prompt:

```
Follow the given rule to solve the question.
Rule:

def add_digits(self, num: int) -> int:
    while num > 9:
        sum = 0
        while num:
            sum += num % 10
            num //= 10
        num = sum
    return num

Q: Given an integer number 15, repeatedly add up all its digits until the result has only one
digit.
```

Response:

1. Initialize
num = 15
2. Outer loop

```python
while num > 9:
```

num = 15
enter the loop
2.1 One iteration
sum = 0
3. Inner loop

```python
while num:
```

num = 15
enter the loop
3.1 One iteration

```python
sum += num % 10
num //= 10
```

num = 15
num % 10 = 5
sum = 0
now,
sum = 0 + 5 = 5
num = 15 // 10 = 1

```python
while num:
```

num = 1
enter the loop
3.1 One iteration

```python
sum += num % 10
num //= 10
```

num = 1
num % 10 = 1
sum = 5
now,
sum = 5 + 1 = 6
num = 1 // 10 = 0

```python
while num:
```

num = 0
do not enter

```python
num = sum
```

sum = 6
now,
num = 6

```python
while num > 9:
```

num = 6
do not enter
4. Return

```python
return num
```

num = 6
So the answer is 6

## G.2 Synthetic Data Generation

22 manually-written code snippets are as below:

```
0: "if list1:\n    list1[-1] += {}",
1: "if list1:\n    list1[0] += {}",
2: "if list1:\n    var = list1.pop(0)\n    list2.append(var)",
3: "if list1:\n    var = list1.pop()\n    list2.append(var)",
4: "list1.insert(0, {})",
5: "list1.sort()",
6: "list1.reverse()",
7: "list1 = list1[1:] if list1 else list1",
8: "if list2:\n    list1.insert(0, list2[0])\nelse:\n    list1.insert(0, {})",
9: "val = list2[-1] if list2 else {}\nlist1.append(val)",
10: "if list1 and list2 and list1[0] > list2[0]:\n    list1.pop(0)",
11: "if list1 and list2 and list1[-1] < list2[-1]:\n    list1.pop()",
12: "if list1:\n    list1.pop(0)",
13: "if list1 and list2:\n    list1.append(list2.pop())",
14: "if list1 and list1[0] % 2 == 0:\n    list1.pop(0)",
15: "if list1 and list1[0] % 2 == 1:\n    list1.pop(0)",
16: "if len(list1) > len(list2):\n    list2.insert(0, list1.pop())",
17: "if list1 and list1[0] > {}:\n    list1.pop(0)",
18: "if list1 and list1[0] < {}:\n    list1.pop(0)",
19: "if list2:\n    list1.append(list2.pop(0))",
20: "if list1:\n    list1.pop()",
21: "if list2:\n    list2.pop()"
```

To further enhance rule diversity, we replace `list1` and `list2` with meaningless random strings in each sampled snippet. Here is a prompt of synthetic data sample with one-shot example:

Here is 1 example:
Follow the given rule to solve the question.
rule:

```python
def process_list(ywhm, erep):
    while ywhm and erep:
        if ywhm:
            ywhm.pop()
        erep = erep[1:] if erep else erep
        if erep and erep[0] % 2 == 0:
            erep.pop(0)
        if erep and erep[0] % 2 == 1:
            erep.pop(0)
        if erep:
            erep.pop(0)
        if ywhm:
            ywhm.pop()
        val = erep[-1] if erep else 53
        ywhm.append(val)
        if erep:
            var = erep.pop(0)
            ywhm.append(var)
    return ywhm
```

Q: Given two lists, ywhm = [3] and erep = [50, 31], what is the final value of ywhm?

1 Initialize
ywhm = [3]

```
erep = [50, 31]
```
2 Main loop

```python
while ywhm and erep:
```

```
ywhm = [3]
erep = [50, 31]
enter the loop
```
2.1 One iteration:

```python
if ywhm:
    ywhm.pop()
```

```
ywhm = [3]
enter if
now,
ywhm = []
```

```python
erep = erep[1:] if erep else erep
```

```
erep = [50, 31]
now,
erep = [31]
```

```python
if erep and erep[0] % 2 == 0:
    erep.pop(0)
```

```
erep = [31]
erep[0] % 2 = 31 % 2 != 0
do not enter if
```

```python
if erep and erep[0] % 2 == 1:
    erep.pop(0)
```

```
erep = [31]
erep[0] % 2 = 31 % 2 == 1
enter if
now,
erep = []
```

```python
if erep:
    erep.pop(0)
```

```
erep = []
do not enter if
```

```python
if ywhm:
    ywhm.pop()
```

```
ywhm = []
do not enter if
```

```python
val = erep[-1] if erep else 53
ywhm.append(val)
```

```
erep = []
val = 53
ywhm = []
now,
ywhm = [53]
```

```python
if erep:
    var = erep.pop(0)
    ywhm.append(var)
```

```
erep = []
do not enter if
```

```python
while ywhm and erep:
```

ywhm = [53]
erep = []
do not enter

```python
return ywhm
```

So the answer is [53]

Follow the above examples to answer the following question:
rule:

```python
def process_list(cybez, eonx):
    while cybez and eonx:
        eonx.reverse()
        if eonx:
            cybez.insert(0, eonx[0])
        else:
            cybez.insert(0, 63)
        if cybez and cybez[0] < 9:
            cybez.pop(0)
        if eonx:
            eonx[-1] += 9
        if cybez and cybez[0] % 2 == 1:
            cybez.pop(0)
        if eonx:
            eonx[0] += 96
        if cybez and cybez[0] % 2 == 0:
            cybez.pop(0)
        if cybez:
            cybez.pop()
    return cybez
```

Q: Given two lists, cybez = [31, 22, 95, 74, 70, 55, 39, 22, 77] and eonx = [53, 70, 50, 28, 15, 81, 92, 6], what is the final value of cybez?

## G.3 Natural Language Rules

To generate rule-following output in natural language, we first convert code-formatted rules into natural language descriptions. The remaining process is the same as generating the output for the code rule. To ensure clarity and avoid ambiguity, each line is prefixed with a number. An example of a rule-following output in natural language is provided below.

Prompt:

Follow the given rule to solve the question.
Rule:

```
1. Begin outer loop:
    1.1 Check whether the number is a single-digit number. If it is
    ↪   not, enter the outer loop; otherwise, the loop is over, go to
    ↪   step (2).
    1.2 One iteration:
        1.2.1 Initialize the sum of the digits of the number in this
        ↪   iteration as sum equals 0.
        1.2.2 Begin inner loop to calculate the digit sum:
```

```
          1.2.2.1 Check whether the number is 0. If the number is
          ↪   not 0, enter the inner loop; otherwise, the loop is
          ↪   over, go to step (1.2.3).
          1.2.2.2 One iteration:
              1.2.2.2.1 Find the last digit of the number and add it
              ↪   to sum.
              1.2.2.2.2 Remove the last digit from the number.
          1.2.2.3 Return to the start of the inner loop.
      1.2.3 After calculating the sum of all digits, update the
      ↪   number with the value of sum.
   1.3 Return to the start of the outer loop.
2. Return the remaining single-digit number.
```

Q: Given an integer number 32, repeatedly add up all its digits until the result has only one digit.

Response:

The number is 32. Begin the process.

```
1. Begin outer loop:
1.1 Check whether the number is a single-digit number. If it is not,
↪   enter the outer loop; otherwise, the loop is over, go to step (2).
```

The number is 32, which is not a single-digit number. Enter the outer loop.

```
1.2 One iteration:
1.2.1 Initialize the sum of the digits of the number in this iteration
↪   as sum equals 0.
```

Initialize sum = 0.

```
1.2.2 Begin inner loop to calculate the digit sum:
1.2.2.1 Check whether the number is 0. If the number is not 0, enter
↪   the inner loop; otherwise, the loop is over, go to step (1.2.3).
```

The number is 32, which is not 0. Enter the inner loop.

```
1.2.2.2 One iteration:
1.2.2.2.1 Find the last digit of the number and add it to sum.
```

number = 32, the last digit of the number is 2. Add it to sum, sum = 0 + 2 = 2.

```
1.2.2.2.2 Remove the last digit from the number.
```

Remove the last digit from the number, number = 32 // 10 = 3.

```
1.2.2.3 Return to the start of the inner loop.
```

Back to the start of the inner loop.

```
1.2.2 Begin inner loop to calculate the digit sum:
1.2.2.1 Check whether the number is 0. If the number is not 0, enter
↪   the inner loop; otherwise, the loop is over, go to step (1.2.3).
```

The number is 3, which is not 0. Enter the inner loop.

```
1.2.2.2 One iteration:
1.2.2.2.1 Find the last digit of the number and add it to sum.
```

number = 3, the last digit of the number is 3. Add it to sum, sum = 2 + 3 = 5.

```
1.2.2.2.2 Remove the last digit from the number.
```

Remove the last digit from the number, number = 3 // 10 = 0.

```
1.2.2.3 Return to the start of the inner loop.
```

Back to the start of the inner loop.

```
1.2.2 Begin inner loop to calculate the digit sum:
1.2.2.1 Check whether the number is 0. If the number is not 0, enter
↪   the inner loop; otherwise, the loop is over, go to step (1.2.3).
```

The number is 0. The loop is over. Go to step (1.2.3).

```
1.2.3 After calculating the sum of all digits, update the number with
↪   the value of sum.
```

sum = 5. Update the number with the value of sum, so number is 5 now.

```
1.3 Return to the start of the outer loop.
```

Back to the start of the outer loop.

```
1. Begin outer loop:
1.1 Check whether the number is a single-digit number. If it is not,
↪   enter the outer loop; otherwise, the loop is over, go to step (2).
```

The number is 5, which is a single-digit number. The loop is over. Go to step (2).

```
2. Return the remaining single-digit number.
```

The number is 5. So the answer is 5.

# H Details of All 86 Tasks

| Split | Domain | Task Name | Description | Length Definition | Reference URL |
|---|---|---|---|---|---|
| RF-pretrain | LeetCode | lc_2582 | We are passing a pillow back and forth along a line of n people for certain time, returning the final holder's position after directional changes at each end. | the number of people | `https://leetcode.com/problems/pass-the-pillow/description/` |
| RF-pretrain | LeetCode | lc_2129 | If the length of the word is 1 or 2 letters, change all letters to lowercase. If the length of the word is 3 or more letters, change the first letter to uppercase and the rest to lowercase. | the number of characters in the word | `https://leetcode.com/problems/capitalize-the-title/description/` |
| RF-pretrain | LeetCode | lc_2210 | Given a 0-indexed integer array `nums`, find out the number of hills and valleys in the array. An index i is part of a hill in nums if the closest non-equal neighbors of i are smaller than nums[i]. Similarly, an index i is part of a valley in nums if the closest non-equal neighbors of i are larger than nums[i] | the length of the array | `https://leetcode.com/problems/count-hills-and-valleys-in-an-array/description/` |
| RF-pretrain | LeetCode | lc_824 | If a word begins with a vowel, append "ma" to the end of the word.If a word begins with a consonant (i.e., not a vowel), remove the first letter and append it to the end, then add "ma". Add one letter "a" to the end of each word per its word index in the sentence, starting with 1. What's the final sentence representing the conversion from sentence to Goat Latin? | the number of words | `https://leetcode.com/problems/goat-latin/description/` |
| RF-pretrain | LeetCode | lc_2103 | Given a string rings of length 2n that describes the n rings that are placed onto the rods. Every two characters in rings forms a color-position pair that is used to describe each ring. How many are the number of rods that have all three colors of rings on them? | the number of rings | `https://leetcode.com/problems/rings-and-rods/description/` |

*Continue on next page...*

| Split | Domain | Task Name | Description | Length Definition | Reference URL |
|---|---|---|---|---|---|
| RF-pretrain | LeetCode | lc_2899 | Given an integer array nums where nums[i] is either a positive integer or -1. We need to find for each -1 the respective positive integer, which we call the last visited integer. To achieve this goal, let's define two empty arrays: seen and ans. Start iterating from the beginning of the array nums.
• If a positive integer is encountered, prepend it to the front of seen.
• If -1 is encountered, let $k$ be the number of consecutive -1s seen so far (including the current -1),
  – If $k \leq$ length of seen, append the $k$-th element of seen to ans.
  – If $k >$ length of seen, append -1 to ans.
Return the array ans. | the length of the array | https://leetcode.com/problems/last-visited-integers/description/ |
| RF-pretrain | LeetCode | lc_2833 | Find the furthest point from origin after given moves. | the number of moves | https://leetcode.com/problems/furthest-point-from-origin/description/ |
| RF-pretrain | LeetCode | lc_2057 | Given a 0-indexed integer array nums, What's the smallest index i of nums such that i mod 10 == nums[i]? | the length of the array | https://leetcode.com/problems/smallest-index-with-equal-value/ |
| RF-pretrain | LeetCode | lc_953 | Given a sequence of words written in the alien language, and the order of the alphabet, return true if and only if the given words are sorted lexicographically in this alien language. | the number of words | https://leetcode.com/problems/verifying-an-alien-dictionary/description/ |
| RF-pretrain | LeetCode | lc_2785 | Given a 0-indexed string s, permute s to get a new string t such that: All consonants remain in their original places. The vowels must be sorted in the nondecreasing order of their ASCII values. | the length of the string | https://leetcode.com/problems/sort-vowels-in-a-string/ |
| RF-pretrain | LeetCode | lc_2460 | Conduct specific operations to an array: perform sequential operations on the array to merge adjacent equal elements by doubling one and zeroing the other, then moving all zeros to the end. | the length of the array | https://leetcode.com/problems/apply-operations-to-an-array/description/ |

*Continue on next page...*

| Split | Domain | Task Name | Description | Length Definition | Reference URL |
|-------|--------|-----------|-------------|-------------------|---------------|
| RF-pretrain | LeetCode | lc_2682 | There are n friends, sitting in a circle and numbered from 1 to n in clockwise order, playing the sircular game. Start at 1st friend and end at 1st friend receive the ball again. What is the serial number of the friend who hasn't caught the ball? | the number of people is given by: ``` n = random.choice( range(1,length)) ``` | https://leetcode.com/problems/find-the-losers-of-the-circular-game/description/ |
| RF-pretrain | LeetCode | lc_1694 | Reformat the given phone number into right format. | the number of characters of the phone number string | https://leetcode.com/problems/reformat-phone-number/description/ |
| RF-pretrain | LeetCode | lc_890 | Given a list of strings words and a string pattern, return a list of words[i] that match pattern. You may return the answer in any order. | the number of words | https://leetcode.com/problems/find-and-replace-pattern/description/ |
| RF-pretrain | LeetCode | lc_2390 | Given a string s containing lowercase English letters and "*", return the string obtained by removing all "*" and the character that comes before "*". | the length of the string | https://leetcode.com/problems/removing-stars-from-a-string/ |
| RF-pretrain | LeetCode | lc_2418 | A list of names and heights are given.Figure out the order of names by their heights. | the number of people | https://leetcode.com/problems/sort-the-people/description/ |
| RF-pretrain | LeetCode | lc_1909 | Given an array nums. Can we remove one element to make it increasing? | length of the array | https://leetcode.com/problems/remove-one-element-to-make-the-array-strictly-increasing/description/ |
| RF-pretrain | LeetCode | lc_1704 | Decide whether the number of vowels in the first half of the string is equal to the number of vowels in the second half of the string. | the length of half of the string | https://leetcode.com/problems/determine-if-string-halves-are-alike/description |

*Continue on next page...*

| Split | Domain | Task Name | Description | Length Definition | Reference URL |
|---|---|---|---|---|---|
| RF-pretrain | LeetCode | lc_1823 | Try to find the winner of the circular game. Rule: the $k$th person next to the start person will be kicked off the game. Find the last person left in the game. | the number of people | https://leetcode.com/problems/find-the-winner-of-the-circular-game/ |
| RF-pretrain | LeetCode | lc_2810 | Your laptop keyboard is faulty, and whenever you type a character 'i' on it, it reverses the string that you have written. Typing other characters works as expected. You are given a 0-indexed string s, and you type each character of s using your faulty keyboard. Return the final string that will be present on your laptop screen. | the number of characters in the string (we assure there is only one "i" in the string) | https://leetcode.com/problems/faulty-keyboard/description/ |
| RF-pretrain | LeetCode | lc_2645 | Given a string word to which you can insert letters "a", "b" or "c" anywhere and any number of times, return the minimum number of letters that must be inserted so that word becomes valid. A string is called valid if it can be formed by concatenating the string "abc" several times. | the length of the string | https://leetcode.com/problems/minimum-additions-to-make-valid-string/ |
| RF-pretrain | LeetCode | lc_2609 | Find the longest balanced substring in a given string. A substring of s is considered balanced if all zeroes are before ones and the number of zeroes is equal to the number of ones inside the substring. Notice that the empty substring is considered a balanced substring. | the length of the string | https://leetcode.com/problems/find-the-longest-balanced-substring-of-a-binary-string/description/ |
| RF-pretrain | LeetCode | lc_2423 | Return true if it is possible to remove one letter so that the frequency of all letters in word is equal, and false otherwise. | the length of the string | https://leetcode.com/problems/remove-letter-to-equalize-frequency/description/ |
| RF-pretrain | LeetCode | lc_2185 | Return the words that start with given prefix. | the number of words | https://leetcode.com/problems/counting-words-with-a-given-prefix/description/ |
| RF-pretrain | LeetCode | lc_2496 | Given an array `strs` of alphanumeric strings, return the maximum value of any string in strs by referring some specific rule. | the length of the array | https://leetcode.com/problems/maximum-value-of-a-string-in-an-array/description/ |

| Split | Domain | Task Name | Description | Length Definition | Reference URL |
|---|---|---|---|---|---|
| RF-pretrain | LeetCode | lc_1275 | A play Tic Tac Toe Game with B. A started first. Given the moves, judge who is the winner of the Tic Tac Toe Game. | the number of moves | https://leetcode.com/problems/find-winner-on-a-tic-tac-toe-game/description/ |
| RF-pretrain | LeetCode | lc_1576 | Given a string s containing only English letters and the "?" character. Convert all the "?" characters into letters such that the final string does not contain any consecutive repeating characters. | the length of the string | https://leetcode.com/problems/replace-all-s-to-avoid-consecutive-repeating-characters/description/ |
| RF-pretrain | LeetCode | lc_1041 | On an infinite plane, a robot initially stands at (0, 0) and faces north."G": go straight 1 unit. "L": turn 90 degrees to the left. "R": turn 90 degrees to the right. The robot performs the instructions given in order, and repeats them forever. Return True if and only if there exists a circle in the plane such that the robot never leaves the circle. | the number of instructions | https://leetcode.com/problems/robot-bounded-in-circle/description/ |
| RF-pretrain | LeetCode | lc_2078 | There are n houses evenly lined up on the street, and each house is beautifully painted. You are given a 0-indexed integer array colors of length n, where colors[i] represents the color of the ith house. Return the maximum distance between two houses with different colors. The distance between the ith and jth houses is abs(i - j), where abs(x) is the absolute value of x. | the number of houses | https://leetcode.com/problems/two-furthest-houses-with-different-colors/description/ |
| RF-pretrain | LeetCode | lc_2016 | Given a 0-indexed integer array nums of size n, find the maximum difference between nums[i] and nums[j] (i.e., nums[j] - nums[i]), such that $0 <= i < j < n$ and nums[i] < nums[j]. | the length of the array | https://leetcode.com/problems/maximum-difference-between-increasing-elements/description/ |
| RF-pretrain | LeetCode | lc_3271 | Conduct hash transformation on the given string. | the length of the string = length * random.randint(2,4) | https://leetcode.com/problems/hash-divided-string/description/ |

*Continue on next page...*

| Split | Domain | Task Name | Description | Length Definition | Reference URL |
|---|---|---|---|---|---|
| RF-pretrain | LeetCode | lc_2529 | Given an array nums sorted in non-decreasing order, find the maximum between the number of positive integers and the number of negative integers. | the length of the array | https://leetcode.com/problems/maximum-count-of-positive-integer-and-negative-integer/description/ |
| RF-pretrain | LeetCode | lc_2047 | Given a string sentence. What is the number of valid words? | the number of words | https://leetcode.com/problems/number-of-valid-words-in-a-sentence/description/ |
| RF-pretrain | LeetCode | lc_2828 | Dtermine whether a string is an acronym of given words | the number of words | https://leetcode.com/problems/check-if-a-string-is-an-acronym-of-words/description/ |
| RF-pretrain | LeetCode | lc_2404 | Find out the most frequent even element in the given array. | the length of the array | https://leetcode.com/problems/most-frequent-even-element/description/ |
| RF-pretrain | LeetCode | lc_2678 | Given a 0-indexed array of strings details. Each element of details provides information about a given passenger compressed into a string of length 15. The eleventh and twelfth digits represent the ages of the person. What is the number of the olders? | the length of the array | https://leetcode.com/problems/number-of-senior-citizens/description/ |
| RF-pretrain | LeetCode | lc_674 | Return the length of the longest continuous increasing subsequence. | the length of the array | https://leetcode.com/problems/longest-continuous-increasing-subsequence/description/ |
| RF-pretrain | LeetCode | lc_605 | Given an integer array flowerbed containing 0's and 1's, where 0 means empty and 1 means not empty, and an integer n, return true if n new flowers can be planted in the flowerbed without violating the no-adjacent-flowers rule and false otherwise. | the length of the array | https://leetcode.com/problems/can-place-flowers/description/ |

*Continue on next page...*

| Split | Domain | Task Name | Description | Length Definition | Reference URL |
|---|---|---|---|---|---|
| RF-pretrain | LeetCode | lc_290 | Given a pattern and a string s, find if s follows the same pattern. (e.g., pattern = "abba", s = "dog cat cat dog" → true) | the number of words in the string | https://leetcode.com/problems/word-pattern/description/ |
| RF-pretrain | LeetCode | lc_414 | Given an integer array nums, return the third distinct maximum number in this array. If the third maximum does not exist, return the maximum number. | the length of the array | https://leetcode.com/problems/third-maximum-number/description/ |
| RF-pretrain | LeetCode | lc_388 | What's the length of the longest absolute path to a file in the abstracted file system? If there is no file in the system, return 0. | the depth of the file system | https://leetcode.com/problems/longest-absolute-file-path/ |
| RF-pretrain | LeetCode | lc_434 | Given a string s, What's the number of segments in the string? A segment is defined to be a contiguous sequence of non-space characters. | the number of segments | https://leetcode.com/problems/number-of-segments-in-a-string/description/ |
| RF-pretrain | LeetCode | lc_228 | Give a sorted unique integer array, return the smallest sorted list of ranges that cover all the numbers in the array exactly,and no extra integer being covered. (e.g., nums = [0,1,2,4,5,7] → ["0→2","4→5","7"] | first generate an array of given length, then remove the repeated element | https://leetcode.com/problems/summary-ranges/description/ |
| RF-pretrain | LeetCode | lc_448 | Given an array nums of n integers where nums is a permutation of the numbers in the range [1, n], return an array of all the integers in the range [1, n] that do not appear in nums. | the length of the array | https://leetcode.com/problems/find-all-numbers-disappeared-in-an-array/description/ |
| RF-pretrain | LeetCode | lc_242 | Determine whether the one word can be converted into another word through alphabetical order adjustment | the number of letters of the word | https://leetcode.com/problems/valid-anagram/description/ |
| RF-pretrain | LeetCode | lc_268 | Given an array nums containing n distinct numbers in the range [0, n], return the only number in the range that is missing from the array. | length = n + 1 | https://leetcode.com/problems/missing-number/description/ |

*Continue on next page...*

| Split | Domain | Task Name | Description | Length Definition | Reference URL |
|---|---|---|---|---|---|
| RF-pretrain | LeetCode | lc_383 | Given two strings `ransomNote` and `magazine`, return true if `ransomNote` can be constructed by using the letters from `magazine` and false otherwise. Each letter in magazine can only be used once in ransomNote. | the length of the string `magazine` | https://leetcode.com/problems/ransom-note/description/ |
| RF-pretrain | LeetCode | lc_682 | You are keeping the scores for a baseball game with strange rules. At the beginning of the game, you start with an empty record. You are given a list of strings operations, where operations[i] is the ith operation you must apply to the record. | the number of operations | https://leetcode.com/problems/baseball-game/description/ |
| RF-pretrain | LeetCode | lc_387 | Given a string s, find the first non-repeating character in it and return its index. If it does not exist, return -1. | the length of the string | https://leetcode.com/problems/first-unique-character-in-a-string/description/ |
| RF-pretrain | LeetCode | lc_345 | Given a string s, reverse only all the vowels in the string and return it. And the vowels can appear in both lower and upper cases, more than once. | the length of the string | https://leetcode.com/problems/reverse-vowels-of-a-string/description/ |
| RF-pretrain | LeetCode | lc_392 | Given two strings `s` and `t`, return true if `s` is a subsequence of `t`, or false otherwise. | the length of string `t` | https://leetcode.com/problems/is-subsequence/description/ |
| RF-pretrain | LeetCode | lc_705 | Design a HashSet without using any built-in hash table libraries. Given operations, return a list of result of each step. | the number of operations | https://leetcode.com/problems/design-hashset/description/ |
| RF-pretrain | LeetCode | lc_796 | Given two strings s and goal, return true if and only if s can become goal after some number of shifts on s. A shift on s consists of moving the leftmost character of s to the rightmost position. | the length of the string s | https://leetcode.com/problems/rotate-string/description/ |
| RF-pretrain | LeetCode | lc_2562 | Given an integer array, compute the concatenation value by repeatedly adding the concatenation of the first and last elements (or the single remaining element) until the array is empty, then returning the total value. | the length of the array | https://leetcode.com/problems/find-the-array-concatenation-value/description/ |

*Continue on next page...*

| Split | Domain | Task Name | Description | Length Definition | Reference URL |
|---|---|---|---|---|---|
| RF-pretrain | LeetCode | lc_1417 | Given an alphanumeric string s (containing lowercase letters and digits), rearrange it such that no two adjacent characters are of the same type (no two letters or two digits in a row). Return the reformatted string if possible; otherwise, return an empty string. | the length of the string | https://leetcode.com/problems/reformat-the-string/description/ |
| RF-pretrain | LeetCode | lc_520 | Given a string word, return true if the usage of capitals in it is right. | the length of the string | https://leetcode.com/problems/detect-capital/description/ |
| RF-pretrain | LeetCode | lc_557 | Given a string s, reverse the order of characters in each word within a sentence while still preserving whitespace and initial word order. | the length of the string | https://leetcode.com/problems/reverse-words-in-a-string-iii/description/ |
| RF-pretrain | LeetCode | lc_541 | Given a string s and an integer k, reverse the first k characters for every 2k characters counting from the start of the string. | the length of the string | https://leetcode.com/problems/reverse-string-ii/description/ |
| RF-pretrain | LeetCode | lc_485 | Given a binary array nums, return the maximum number of consecutive 1's in the array. | the length of the array | https://leetcode.com/problems/max-consecutive-ones/description/ |
| RF-pretrain | LeetCode | lc_344 | Write a function that reverses a string. The input string is given as an array of characters s. | the length of the string | https://leetcode.com/problems/reverse-string/description/ |
| RF-pretrain | LeetCode | lc_500 | Given an array of strings words, return the words that can be typed using letters of the alphabet on only one row of American keyboard . | the number of words | https://leetcode.com/problems/keyboard-row/description/ |
| RF-pretrain | LeetCode | lc_482 | Reformat the given license key string s by removing all dashes, converting letters to uppercase, and grouping the characters into segments of length k (except possibly the first group), separated by dashes. | the length of the string | https://leetcode.com/problems/license-key-formatting/description/ |
| RF-pretrain | LeetCode | lc_896 | An array is monotonic if it is either monotone increasing or monotone decreasing.Given an integer array nums, return true if the given array is monotonic, or false otherwise. | the length of the array | ttps://leetcode.com/problems/monotonic-array/description/ |

*Continue on next page...*

| Split | Domain | Task Name | Description | Length Definition | Reference URL |
|-------|--------|-----------|-------------|-------------------|---------------|
| RF-pretrain | LeetCode | lc_551 | Given a string s representing an attendance record for a student where each character signifies whether the student was absent, late, or present on that day. Return true if the student is eligible for an attendance award, or false otherwise. | the length of the string | https://leetcode.com/problems/student-attendance-record-i/description/ |
| RF-pretrain | LeetCode | lc_1556 | Given an integer n, add a dot (".") as the thousands separator and return it in string format. | the number is given by ``` random.randint( 1000, 1000+ 10**length) ``` | https://leetcode.com/problems/thousand-separator/description/ |
| RF-pretrain | LeetCode | lc_2869 | You are given an array nums of positive integers and an integer k. In one operation, you can remove the last element of the array and add it to your collection. Return the minimum number of operations needed to collect elements 1, 2, ..., k. | the array and k is given by: ``` nums = random.sample([i for i in range(1, length+1)]*2, k=int(length*1.9)) k = random.randint(3, length) ``` | https://leetcode.com/problems/minimum-operations-to-collect-elements/description/ |
| downstream | LeetCode | lc_258 (Add Digits) | Given an integer, repeatedly sum its digits until the result is a single digit. | the number of digits in the integer | https://leetcode.com/problems/add-digits/description/ |
| downstream | LeetCode | lc_283 (Move Zeros) | Given a list of integers, move all zeros to the end while preserving the relative order of the non-zero elements. | the length of the list | https://leetcode.com/problems/move-zeroes/description/ |
| downstream | LeetCode | lc_125 (Valid Palindrome) | Given a string s, return true if it is a palindrome after removing all non-alphanumeric characters and converting it to lowercase; otherwise, return false. | half length of the string | https://leetcode.com/problems/valid-palindrome/description/ |
| downstream | LeetCode | lc_2544 (Alternate Digit Sum) | Given a positive integer where the most significant digit has a positive sign, and each subsequent digit has the opposite sign of its adjacent digit, return the sum of these signed digits. | the number of digits in the integer | https://leetcode.com/problems/alternating-digit-sum/description/ |

*Continue on next page...*

| Split | Domain | Task Name | Description | Length Definition | Reference URL |
|---|---|---|---|---|---|
| downstream | LeetCode | lc_1598 (Crawler Log Folder) | Determine the final folder after performing the operations in the given list, where ../ moves up one level, ./ stays in the current folder, and x/ moves into folder x. | the number of operations | https://leetcode.com/problems/crawler-log-folder/description/ |
| downstream | LeetCode | lc_3324 (String Sequence) | Given a target string, return a list of all strings that appear on the screen in order, using the minimum key presses. Key 1 appends "a" to the string, and Key 2 changes the last character to its next letter in the alphabet. | The sum of each letter's ASCII code minus 96 in the target string should equal its length. For example: given input 'abc', the length is 1+2+3=6. | https://leetcode.com/problems/find-the-sequence-of-strings-appeared-on-the-screen/ |
| downstream | LeetCode | lc_2677 (Chunk Array) | Given array and chunk size, split the array into subarrays of a given size. | the length of the array | https://leetcode.com/problems/chunk-array/description/ |
| downstream | LeetCode | lc_461 (Hamming Distance) | The Hamming distance between two integers is the number of positions at which the corresponding bits are different. Given two integers in binary representation, return their Hamming distance. | The bit length of the integers | https://leetcode.com/problems/hamming-distance/description/ |
| downstream | NUPA | Get Digit Integer | Given a number and an integer i, return the i-th digit. | the number of digits in the given number | https://arxiv.org/abs/2411.03766 |
| downstream | NUPA | Add Integer | Add the two given integers together. | the maximum of the number of digits of the two given numbers | https://arxiv.org/abs/2411.03766 |
| downstream | NUPA | Digit Max Integer | Compare two numbers digit by digit and return the larger digit at each position, treating any missing digits as 0. | the number of digits in the given number | https://arxiv.org/abs/2411.03766 |
| downstream | NUPA | Length Integer | Return the total length (i.e., the number of digits) of a number. | the number of digits in the given number | https://arxiv.org/abs/2411.03766 |
| RF-pretrain | BBH | Dyck Languages | Determine whether a given sequence of parentheses forms a valid, properly nested structure according to Dyck language rules. | the number of bracket pairs | https://arxiv.org/abs/2210.09261 |

*Continue on next page...*

| Split | Domain | Task Name | Description | Length Definition | Reference URL |
|---|---|---|---|---|---|
| RF-pretrain | BBH | Hyperbaton | Given a sentence with scrambled adjectives, determine whether their current order follows grammatical rules. | the number of adjectives | https://arxiv.org/abs/2210.09261 |
| RF-pretrain | BBH | Navigate | Follow a set of directional instructions to determine the final position. | the number of instructions | https://arxiv.org/abs/2210.09261 |
| RF-pretrain | BBH | Object Counting | Accurately count the number of specified objects. | the number of given objects | https://arxiv.org/abs/2210.09261 |
| RF-pretrain | BBH | Reverse List | Given a python list, return it in the exact opposite order. | the length of the list | https://arxiv.org/abs/2210.09261 |
| RF-pretrain | BBH | Word Sorting | Sort a given list of words in strict dictionary order. | the number of words | https://arxiv.org/abs/2210.09261 |
| RF-pretrain | Symbolic Reasoning | Coin Flip | Given a series of operations, answer whether a coin is still heads up after people either flip or don't flip the coin. (e.g., "A coin is heads up. Phoebe flips the coin. Osvaldo does not flip the coin. Is the coin still heads up?" → "no") | the number of people who flip the coin | https://arxiv.org/abs/2201.11903 |
| RF-pretrain | Symbolic Reasoning | Last Letter Concatenation | Given a list of words, concatenate the last letters of each word and return the string. (e.g., "Amy Brown" → "yn") | the number of words | https://arxiv.org/abs/2201.11903 |

