# OpenReview forum: "Beyond Single-Task: Robust Multi-Task Length Generalization for LLMs"
_NeurIPS.cc/2025/Conference — NeurIPS 2025 poster_

### Official Review · Reviewer_MTxe · 2025-06-30

**Clarity:** 3
**Significance:** 2
**Originality:** 2
**Rating:** 3
**Confidence:** 4

**Summary:**

This paper studies the problem of length generalization of LLM in multi-task setting, they introduce Meta-RFFT—an extension of RFFT that teaches a model to internalise general, code-style rules across 74 diverse tasks and then reuse that know-how on completely new problems. After this rule-following “pre-training”, a 32-billion-parameter model needs only a handful of short examples—or even a single in-context demonstration—to solve much longer versions of unseen tasks with near-perfect accuracy. Error analysis shows the win comes largely from mastering loop control, a transferable primitive that vanilla, single-task fine-tuning never really learns.

**Questions:**

Refer to weaknesses

**Ethical Concerns:**

["NO or VERY MINOR ethics concerns only"]

**Final Justification:**

After reading the authors’ rebuttal and engaging in a thorough discussion, I believe my original concerns regarding W3 and W4 have been resolved.

However, I am still slightly leaning towards rejection, as W1 and W2 persist even after discussions with the authors. Specifically:
(i) in my view, the contribution compared to the most relevant work [1] is not significant enough for a NeurIPS publication; and (ii) there is a lack of a systematic and scientific approach for constructing datasets for cross-task length generalization—the principles argued by the authors remain vague and subjective.

Lastly, I feel the scope of this paper aligns more closely with the D&B track rather than the main track.

[1] Case-based or rule-based: How do transformers do the math? ICML 2024.

**Limitations:**

N.A.

**Paper Formatting Concerns:**

N.A.

**Quality:**

2

**Strengths And Weaknesses:**

**Strengths:**

1. The problem being studied in this paper is new and interesting.

2. This paper shows strong empirical performances in length generalization compared to baselines.

3. This paper is overall very well written and easy to follow.

4. This paper proposes a data construction pipeline for length generalization tasks, which could be of interest to practitioners.

**Weaknesses:**

1. The contribution, compared to the most relevant work - RFFT, seems incremental; this paper simply generalizes the problem to a multi-task setting, with very few actual technical contributions.

2. This is mainly a heuristic paper and does not introduce any new insights to the problem; some claimed insight, such as learning transferrable patterns is beneficial for multi-task learning, is, in fact, commonly known principles in areas such as Domain Generalization, Multi-task Learning, as well as Instruction Tuning.

3. The authors did not report how Meta-RFFT will hurt the model's performances on other metrics, such as common knowledge, reasoning, and preference alignment. Since the task being considered in this paper is relatively narrow, a significant gain of the length generalization capability at the dispense of other capabilities of the LLM is highly unfavorable.

4. The objective of this study is somewhat confusing to me - are authors trying to improve the out-of-task length generalization ability of the LLM, or are they simply improving the coverage of the fine-tuning dataset? Because it seems that the tasks authors are evaluating are precisely the same as the datasets you constructed, in that case, this paper seems trivial compared to RFFT since it just improves the coverage of the training data and does not truly generalizes to unseen tasks. In real-world practices, you cannot enumerate all possible tasks at the training time, so a sensible evaluation, in my opinion, is to evaluate how generalizable the patterns you learned to the unseen tasks.

---

> ### Author Rebuttal · Authors · 2025-07-31
>
> We sincerely appreciate the reviewer’s constructive feedback. We hope the following responses will fully address your concerns and further clarify our contributions.
>
> # W1:
> We would like to clarify the contribution and novelty of our work as follows:
> 1. **First work to explore cross-task length generalization**
>     * We fully acknowledge that multi-task training is a well-established concept across many domains. However, our work introduces a novel perspective by being the first to explore **cross-task transfer of length generalization** through multi-task training. Length generalization presents a unique challenge due to its inherent **out-of-distribution (OOD)** nature—models are trained on short sequences but must generalize to much longer ones. We show through experiments that traditional multi-task methods like instruction tuning (e.g., Tulu3 in Table 3) fail under such conditions, since the reasoning steps required at test time go far beyond those seen during training.
> 2. **Meta rule-following through multi-task training**
>     * While multi-task training itself is not new, our key insight is that training on a large number of (rule, execution trace) pairs teaches the model to **internalize transferable, rule-based reasoning patterns** through shared computation primitives, such as loop maintainence. This results in a **meta rule-following capability** that supports length generalization across diverse tasks. For example, our RF-pretrained 32B model achieves **95% accuracy on 30-digit addition**, even though it has never seen that task during training—dramatically outperforming much larger models like DeepSeek-R1-671B (72%) and QwQ-32B (32%).
> 3. **Why training on many rules matters**
>     * Humans have developed an extensive array of rules throughout history, encompassing both natural laws and regulatory frameworks. For instance, the rules for integer addition are clearly outlined in textbooks, on wikis, and within computer programs. Although these rules may be present within the large training corpus during pretraining for LLMs, current training methods fail to fully leverage them. The **relationships** between the “rules” and their corresponding “instances” remain **inadequately modeled**. As a result, while recalling rules is relatively straightforward for models, they **struggle to apply these rules** to specific instances. This results in failures to generalize in length on seemingly simple problems. Meta-RFFT addresses this by explicitly linking rules and step-by-step executions across many tasks, encouraging cross-task rule-based reasoning. The idea of modeling the relationship between rules and rule-execution traces is novel beyond RFFT.
> 4. **Key technical contribution beyond RFFT**
>     * We collect 86 diverse tasks spanning code execution, number processing, logic, and symbolic reasoning—far beyond the 3 tasks used in RFFT—and generate 310k rule-following samples for RF-pretraining across 74 of these tasks.
>     * Through this, the model learns a meta rule-following ability: it can solve unseen tasks with only 1-shot prompting, as shown in Section 4.3 (Figures 5 & 9). This enables **in-context transfer** of rule-following behavior, reducing or even **eliminating the need for downstream fine-tuning**—greatly enhancing the practicality and usability of our approach over RFFT.
>     * Besides, the RF-pretrained model demonstrates superior performance in various downstream tasks to vanilla RFFT.
> 5. **Beyond prior length generalization literature**
>     * Previous studies on length generalization focus on **single-task, small-scale** transformers **trained from scratch**. These models, despite generalizing well in certain tasks, have little generic language ability. Our work aligns with LLM's core multi-task advantage and explore length generalization in a **multi-task, post-training** setting, using **pretrained LLMs**.
>
> # W2:
> Indeed, learning transferable patterns aids generalization is a well-established principle in areas such as Domain Generalization and Instruction Tuning, but what are the "transferrable patterns" and how to find them for specific scenarios such as length generalization, matter much more than the principle itself. In this regard, we respectfully argue that our work innovatively introduces a **new insight specific to the problem of length generalization** in large language models: that learning explicit, transferable **rule-following patterns** via multi-task training significantly enhances **OOD generalization on unseen tasks**, which is a direction not addressed in prior work. In other words, our work answers it is the "rules" and their underlying loop structure that make the transfer happen.
>
> By explicitly teaching rules and execution traces across many tasks, Meta-RFFT enables models to internalize meta-level computation strategies, such as loop maintenance, that generalize both across lengths and tasks. This is not merely multi-task learning—it is **the first demonstration of cross-task transfer in length generalization**, enabled by explicit rule-based supervision at scale.
>
> We believe this is a meaningful insight with practical implications for building LLMs that generalize more robustly in reasoning-intensive scenarios.
> # W3:
> We appreciate the reviewer for pointing out this important and practical concern.
>
> Meta-RFFT involves substantial RF-pretraining, which indeed can affect the model’s general capabilities due to distribution shift, known as catastrophic forgetting. But typically this is easily addressed by mixing some pretraining corpus during fine-tuning.
>
> To verify this, we conduct a supplementary experiment. As we don't have access to the pretraining data, we mix GSM8K training data into downstream finetuning alongside the original meta-rule-following data to verify that the reasoning ability on GSM8K can be kept after RFFT. More specifically, for each training example in GSM8K, we use Qwen2.5-7B-Instruct to generate 10 candidate answers and filter the corrrect traces as the training set, forming a filtered GSM8K dataset of 7182 samples.
>
> The results on length generalization (averaged across 8 LeetCode tasks) and GSM8K test accuracy are shown below. We observe that merely mixing 7k samples (compared to 310k Meta-RFFT data) effectively recovers the model’s reasoning performance on GSM8K, while maintaining the strong multi-task length generalization ability. Notably, Meta-RFFT still significantly outperforms vanilla RFFT, demonstrating that **general capabilities can be preserved with minimal additional data**.
>
> |   | ACC_Len30 | Max_Len_90% |
> |---|----|---|
> | vanilla RFFT  | 0.34   | 7.00    |
> | Meta-RFFT    | 0.82    | 22.75   |
> | Meta-RFFT with general data | 0.79      | 21.25       |
>
> |   | GSM8K Test ACC |
> |---|---|
> | Qwen2.5-7B-Instruct  |  87.95%   |
> | Qwen2.5-7B-Instruct (Meta-RFFT with general data) | 87.80%     |
>
> # W4:
> We’d like to clarify a key point: **the evaluation tasks are entirely distinct from the training tasks**, and the focus of this paper is precisely on **generalizing to unseen tasks and unseen lengths**, not merely expanding training coverage. Besides, we conduct additional experiments on a real-world task **the Airline Baggage Cost Estimation task**, which is very different from the tasks in our paper and show the results below.
>
> * As shown in Figure 2, the 12 evaluation tasks (8 from LeetCode, 4 from NUPA in Figure 3) have **no overlap** with the 74 tasks used during RF-pretraining. We will make this clearer in the revision.
> * In particular, the NUPA tasks, used solely for evaluation, are completely held out during RF-pretraining—yet the model generalizes well to these tasks after minimal adaptation.
> * In the downstream adaptation setting, the model is fine-tuned only on lengths 1–5 and then tested on unseen lengths 6–30, to assess true **OOD length generalization**.
> * Beyond that, we also evaluate the model’s ability to handle **unseen tasks via 1-shot in-context learning**, without any fine-tuning. As shown in Figure 5 (7B) and Figure 9 (32B, Appendix E.1), the RF-pretrained model can directly adapt to new tasks with only a single example, achieving strong performance and robust length generalization.
>
> In short, **Meta-RFFT is explicitly designed and evaluated for generalization to tasks and lengths never seen during training**—in stark contrast to simply expanding dataset coverage. We show that it enables LLMs to:
> * (a) **extrapolate to longer input lengths** (length generalization); and
> * (b) **adapt to entirely unseen tasks** through minimal examples (cross-task transfer).
>
> ---
> ## Further Results on Real-World Task
> To validate the practical value of Meta-RFFT beyond symbolic tasks, we additionally conduct experiments on a real-world scenario: **the Airline Baggage Cost Estimation task** from [1].
>
> This task asks the model to compute the total travel cost for passengers according to American Airlines' policy. Solving it **requires understanding and applying natural language regulations, which go beyond what can be fully captured in formal code**. Current LLMs perform poorly on it, and even strong models like Claude-3.5 and o1-preview degrade rapidly with input length (see the table below).
>
> We train models on samples with length 1-8, and report test accuracy on OOD lengths 10-16. As shown below, **Meta-RFFT significantly outperforms vanilla RFFT in the practical domain**:
>
> | length| 2 | 4 | 6 | 8 | 10 | 12 | 14 | 16 |
> |---|---|---|---|---|---|---|---|---|
> | Vanilla RFFT | 0.94 | 0.72 | 0.78 | 0.56 | 0.44 | 0.16 | 0.00 | 0.08 |
> | Meta-RFFT    | 1.00 | 1.00 | 0.92 | 0.90 | 0.74 | 0.38 | 0.24 | 0.26 |
>
> | length  | 5 | 8 | 11|
> |---|---|---|---|
> | Claude-3.5 Sonnet (0-shot) | 0.04 | 0.00 | 0.01 |
> | Claude-3.5 Sonnet (1-shot) | 0.29 | 0.30 | 0.11 |
> | o1-preview (0-shot) | 0.54 | 0.37 | 0.21 |
> | o1-preview (1-shot)| 0.63 | 0.55 | 0.46 |
>
>
> [1] Zhou et al., RULEARENA: A Benchmark for Rule-Guided Reasoning with LLMs in Real-World Scenarios, 2025.

---

> > ### Comment · Reviewer_MTxe · 2025-08-05
> >
> > Thank the authors for their quality review, however, the following concerns presists:
> >
> > > W1 (3)
> >
> > I concur with the authors' premise that leveraging a diverse set of tasks is fundamental to multi-task learning. However, the central weakness remains: the paper does not offer a unified or principled methodology for constructing such a dataset.
> >
> > The current work successfully presents a specific, curated dataset that yields strong empirical results on a predefined set of validation tasks. Yet, it fails to provide a generalizable framework or transferable "philosophical guidance" that would enable other researchers to construct similar high-quality datasets for different domains or goals. The contribution currently reads as, "We created a successful dataset," rather than, "We are proposing a novel and principled method for creating such datasets." The emergent abilities observed appear to be a **post-hoc** discovery resulting from this specific data artifact, not from a novel, principled methodology.
> >
> > > W1 (4)
> >
> > Unfortunately I personally do not view increasing the coverage of the task and datasets as "technical contribution", in fact that is precisely the motivation why there is a seperate track for datasets & benchmarks - it is reserve for papers with notable contribution and significance but relatively less technical contributions. In addition, the observed "meta rule-following", "in-context transfer" abilities are results from improved datasets, but not novel methodlogies or any principled approaches.
> >
> > > W3
> >
> > It's good to see Meta-RFFT does not significantly hurts the performances on GSM8K, however I am looking for results on common knowledge (ARC, MMLU), and preference alignment (Alpaca-Eval, MT-Bench) as well.
> >
> > > W4
> >
> > I thank the authors for clarifying this point - this was indeed a bit sutble, I encourage the authors to highlight this in the experimental setup by clearly listing out what tasks are used for training what are used for validation.
> >
> > In addition, how are the held-out validation tasks determined? It seems that a more rigourous approach would be performing something like cross-validation - randomly splitting a certain ratio of tasks for validation and average over multiple seeds.
> >
> > I realize some of these requests mean extra runs under tight deadlines, but even small proof-of-concepts (partial runs, ablation sketches) would strengthen the argument that the findings generalize.

---

> ### Author Response · Authors · 2025-08-06
> **Response to reviewer MTxe (1/3)**
>
> Thanks for your instructive feedback! We hope that we can address the remaining concerns as follows:
> ## W1 (3) & (4)
> >  The central weakness remains: the paper does not offer a unified or principled methodology for constructing such a dataset.
>
> > The contribution currently reads as, "We created a successful dataset," rather than, "We are proposing a novel and principled method for creating such datasets."
>
> We would like to clarify that the core contribution of our paper is that, we for the first time demonstrate that **length generalization can be transferred across tasks** through the Meta-RFFT method. Constructing this large-scale dataset is a necessary step to prove this claim, but definitely not our whole contribution. Moreover, we have a clear motivation and principled insight to construct such a rule-following dataset for cross-task length generalization transfer. We believe this insight is **beneficial and inspirational for other domains and goals** requiring nontrivial OOD generalization. Below, we detail our logic for constructing this dataset:
> 1. **First, we hope to reach a consensus**: Cross-task length generalization is a **challenging** and **important** problem.
> 2. However, merely expanding training data coverage or standard multi-task learning methods, such as instruction tuning, are **not capable** of achieving **out-of-distribution (OOD) generalization**, as evidenced by our experiments of Tulu3 on length generalization tasks (Table 3 in the original paper).
> 3. Our method, Meta-RFFT, enables **cross-task OOD generalization**. The core reason we have such a belief and thus constructing the necessary dataset is, we observe that tasks requiring infinite generalization on the "length" dimension all need to master a common ability—**"looping"**. But how can we teach models to **master the underlying looping ability** instead of **overfitting the specific rules** for a task (what RFFT does)? Our solution is to manually construct a dataset of **diverse** length-generalization tasks and prepare extensive training data across **different lengths** with **strict (rule, execution) format**. Such extensive data force the model to learn the underlying common structure across tasks, thus mastering the meta ability of looping which allows cross-task length generalization.
> 4. Broadly speaking, we are attempting to model the relationship between **rules and instances**, which is a crucial insight for other domains as well. In OOD scenarios, the key is first identifying **shared meta rules** (looping here) and then fine-tuning the model across diverse tasks sharing such meta rules. This enables the model to perform well on **unseen tasks and unseen complexity**. Humans have developed a vast array of rules throughout history, including both natural laws and regulatory frameworks. For example, the rules for integer addition are clearly laid out in textbooks, wikis, and computer programs. While these rules may be present in the large training corpus used for pretraining LLMs, current training methods fail to fully utilize them, i.e., LLMs can recite them but cannot faithfully execute them. The relationship between these **rules** and their corresponding **instances** is **often inadequately modeled**, which causes models to **struggle to apply these rules** to specific instances, leading to poor generalization on tasks involving length. Meta-RFFT addresses this by explicitly modeling the relationship between rules and rule-execution traces across tasks, fostering **OOD rule-based reasoning**, as evidenced by our impressive one-shot performance on unseen tasks. This is analogous to experienced human learners -- the more tasks they learn, the faster they are capable of learning a new task. Ultimately, human experts can directly **execute a rule they see for the first time**, faithfully across lengths, with few or even zero examples. Such important ability is missing from current LLMs/agents and our work alleviates this gap.
> 5. **Regarding inspiration for other domains**, as shown in the original paper, Meta-RFFT proves **beneficial in downstream tasks that involve natural language rules**, not just formal language rules (Figure 6). This greatly expands the applicability of Meta-RFFT to real-world rule-following domains, such as ***agentic workflow execution*** and ***reading and executing manuals or instructions***. To further validate the method on practical tasks, we conducted additional experiments on the **Airline Baggage Cost Estimation task**, with detailed results shared in the previous response. We believe that the results involving natural language rules and real-world tasks demonstrate that the meta rule-following capability gained by training on vast (rule, execution) pairs can **inspire the construction of similar (rule, execution) pairs for other tasks that require rule-following**, which is a critical capability involving a wide range of practical applications.

---

> ### Author Response · Authors · 2025-08-06
> **Response to reviewer MTxe (2/3)**
>
> ## continuing W1 (3) & (4)
> > The emergent abilities observed appear to be a post-hoc discovery resulting from this specific data artifact, not from a novel, principled methodology.
>
> > the observed "meta rule-following", "in-context transfer" abilities are results from improved datasets, but not novel methodlogies or any principled approaches.
>
> Based on the clarifications above, we would like to emphasize that the creation of the dataset was driven by a **clear insight** and **methodological necessity**. Specifically, we strongly believe that Meta-RFFT is essential for enabling **cross-task length generalization transfer**. The dataset was not created arbitrarily, but rather it was carefully designed following a principle (modeling common structure underlying rule-execution pairs) to **validate** our core claim that length generalization can be transferred across tasks. The observed empirical results were not coincidental, but a **direct consequence** of our principled approach.
>
> ## W3
> > I am looking for results on common knowledge (ARC, MMLU), and preference alignment (Alpaca-Eval, MT-Bench) as well.
>
> Thanks for your constructive advice! We have extended our evaluation to include common knowledge and preference alignment.
>
> Similar to our previous experiments on GSM8K, we mix general data into the downstream fine-tuning alongside the original rule-following data to ensure that general ability is preserved. The results of the model's general ability and length generalization are shown below. Regarding length generalization results, due to tight time constraint, we test on 2 downstream task: `LC Valid Palindrome` and `NUPA Add Integer`. We observe that by mixing 6,872 training samples for ARC and 8,000 for Alpaca, the model’s general ability is effectively restored, while still maintaining strong multi-task length generalization.
>
> Specifically, for Alpaca-Eval, we observe that the model’s win rate improves significantly. This improvement is due to the model generating longer outputs, which is preferred in the win rate calculation. So we add another metric `length control win rate` introduced by [1] to provide a more comprehensive result.
>
> * **General ability results**:
>
> | | Qwen2.5-7B-Instruct | Qwen2.5-7B-Instruct (Meta-RFFT) |
> |------|----|----|
> | ***common knowledge***  | |
> | ARC  | 87.88 | 88.92  |
> | ***preference alignment***  |  |  |
> | Alpaca-Eval (win rate)  | 16.77 | 23.54 |
> | Alpaca-Eval (length controlled win rate)|25.61|23.90|
>
> * **Length generalization results**:
>
> |   | ACC_Len30 (%) | Max_Len_90% |
> |-----|----|----|
> | vanilla RFFT       | 20.0  | 1.0  |
> | Meta-RFFT    | 57.5 |8.0 |
> | Meta-RFFT with ARC |62.5 |8.0 |
> | Meta-RFFT with Alpaca |60.0 |8.0 |
>
> ---
> [1] Dubois et al., Length-Controlled AlpacaEval: A Simple Way to Debias Automatic Evaluators, 2025.

---

> ### Author Response · Authors · 2025-08-06
> **Response to reviewer MTxe (3/3)**
>
> ## W4
> > I encourage the authors to highlight this in the experimental setup by clearly listing out what tasks are used for training what are used for validation.
>
> We apologize for any confusion. In the original paper, we emphasized in Figure 2 that the tasks used during the RF-pretraining phase do not overlap with those used in the downstream adaptation phase. Additionally, Table 1 further illustrates the task distribution across domains for both RF-pretraining and downstream adaptation. We will make sure to highlight this distinction more clearly in the revised version.
>
> > how are the held-out validation tasks determined?
> The held-out validation tasks were determined as follows:
>
> First, we retained the NUPA dataset for validation, as it covers numeric processing tasks, which present a significant domain shift compared to the other three datasets. This makes it an ideal choice for evaluating performance transfer across different task domains.
>
> Secondly, for the LeetCode tasks, we manually selected challenging tasks that also reflect practical real-world scenarios. For example, the `LC Crawler Log Folder` task involves determining the final folder after operations like ../ (move up) and x/ (enter folder), which simulates real-world file system navigation—a common problem in software development. The `LC Hamming Distance` task calculates the difference between two integers' binary representations, which mirrors practical applications in areas such as data comparison, error detection, and cryptography. These tasks demonstrate that our downstream tasks involve reasoning applicable to real-world computational problems.
>
> Finally, for all downstream tasks, we strictly checked—both manually and using LLMs—that there were no data overlaps with the tasks used in the RF-pretraining phase to avoid any potential data contamination.
>
> > It seems that a more rigourous approach would be performing something like cross-validation - randomly splitting a certain ratio of tasks for validation and average over multiple seeds.
>
> Thank you for the suggestion. We agree that a more rigorous approach, such as cross-validation with random task splits and averaging over multiple seeds, would be beneficial. However, due to limited computation resources, we were unable to complete multiple runs of RF-pretraining during the discussion period. That said, we conduct a preliminary evaluation as follows:
>
> We re-run the RF-pretraining **with only LeetCode tasks**. Due to resource constraints, we evaluate the model on **4 downstream tasks from the original paper**: 2 from LeetCode (Move Zeros & Valid Palindrome) and 2 from NUPA (Add Integer & Digit Max Integer). Additionally, we test the model on **3 tasks (2 from Symbolic Reasoning and 1 from Big-Bench Hard), which have been included in the original RF-pretraining but excluded in this new run**. All these tasks are unseen during the RF-pretraining phase.
>
> The results, shown below, indicate a slight performance drop on the tasks that were used in the original paper, likely due to reduced task diversity in this new pretraining run. Nevertheless, **Meta-RFFT still significantly outperforms vanilla RFFT**, further validating the effectiveness of our approach. Furthermore, we test the model trained on LeetCode-only tasks on the Big-Bench Hard and Symbolic Reasoning tasks, and **Meta-RFFT continues to outperform vanilla RFFT**, confirming that our method is robust across tasks.
>
> |  | ACC_Len30 (%) | Max_Len_90% |
> | --- | --- | --- |
> |***results on LC and NUPA***||
> | Meta-RFFT (complete pretrain dataset)| 56.0 |	16.0 |
> | Meta-RFFT (LeetCode only) | 51.0 | 12.5 |
> | Vanilla RFFT | 19.0 | 5.5 |
> |***results on SR and BBH***||
> | Meta-RFFT (LeetCode only) | 68.0 | 20.7 |
> | Vanilla RFFT | 37.0 | 9.3 |
>
> ---
> We are eager to continue addressing any additional questions or concerns you may have. We sincerely appreciate your review and look forward to hearing whether our rebuttal has sufficiently addressed your concerns.

---

> > ### Comment · Reviewer_MTxe · 2025-08-08
> >
> > Thank the authors for their comprehensive rebuttal, I am generally happy with W1 and W4, though I personally still think calling those heuristics a "methodology" is debatable; however, the authors do present some valid points here.
> >
> > In addition, I am particularly curious about W3: we can see academic benchmark performances and preference alignment performances increase after Meta-RFFT, despite the dataset is not related to those tasks at all? I found this to be quite surprising and would very much like to hear authors' further analysis for these results.

---

> > > ### Author Response · Authors · 2025-08-08
> > > **Further Clarifications on W3**
> > >
> > > Thank you for considering our rebuttal comprehensive and valid! We also appreciate your curiosity regarding W3.
> > >
> > > > We can see academic benchmark performances and preference alignment performances increase after Meta-RFFT, despite the dataset not being related to those tasks at all?
> > >
> > > For **ARC**, the observed improvement is modest (**+1.04%**). Given our evaluation temperature of 0.3, this is likely *within normal variance*. To verify this, we additionally run the experiments three times and report the mean and variance in the table below. We also note that the base model *occasionally produces answers in an incorrect format* that can not be parsed (~6/1172=0.51%), whereas the Meta-RFFT model—having undergone SFT—does not exhibit this issue. Overall, this suggests that Meta-RFFT does not degrade common-knowledge performance and such capabilities are easy to retain.
> > >
> > > | | Qwen2.5-7B-Instruct | Qwen2.5-7B-Instruct (Meta-RFFT) |
> > > |------|----|----|
> > > |***ARC***|||
> > > | accuracy (%)  | $87.80\pm 0.14$ | $88.40\pm 0.48\%$  |
> > >
> > > For **Alpaca-Eval**, the `win rate` shows a substantial gain (**+6.77%**). However, as pointed out by [1], *the original win rate metric has a bias toward longer outputs*. Since Meta-RFFT trains the model for length generalization, it tends to produce longer responses, which are favored under the standard metric. We report the average response token num in the table below ($1445\rightarrow 2446$). To control for this bias, we additionally report the `length-controlled win rate` from [1], which *reduces the bias for output length*. Under this more balanced metric, alignment performance shows only a slight decrease (**−1.71%**), which we consider acceptable.
> > >
> > > | | Qwen2.5-7B-Instruct | Qwen2.5-7B-Instruct (Meta-RFFT) |
> > > |------|----|----|
> > > |***Alpaca-Eval***|||
> > > | win rate  | $16.77$ | $23.54$ |
> > > | length control win rate|$25.61$|$23.90$|
> > > | avg response length|$1445$|$2446$|
> > >
> > > In summary, the ARC results indicate that general knowledge ability is well preserved, and the Alpaca-Eval analysis suggests that the apparent large improvement is largely due to length bias, while true preference alignment remains largely unchanged.
> > >
> > > We hope the above explanation addresses your concern. If you have any further questions, we would be more than happy to discuss them. If this clarifies the issue for you, we would sincerely appreciate it if you could consider revising your score. Thank you again for your review and thoughtful feedback.
> > >
> > >
> > > ---
> > > \[1] Dubois et al., Length-Controlled AlpacaEval: A Simple Way to Debias Automatic Evaluators, 2025.

---

> > > > ### Comment · Reviewer_MTxe · 2025-08-08
> > > >
> > > > Thank the authors for their responses, I will adjust my evaluation accordingly.

---

> > > > > ### Author Response · Authors · 2025-08-08
> > > > > **Sincere Thanks for Your Review and Updated Evaluation!**
> > > > >
> > > > > We sincerely thank you for the time and effort you have devoted to reviewing our paper and for providing constructive feedback throughout the process. We are very glad that we were able to address your concerns, and it is great to hear that you have decided to adjust your evaluation accordingly.

---

### Official Review · Reviewer_tbVk · 2025-07-01

**Clarity:** 3
**Significance:** 2
**Originality:** 3
**Rating:** 5
**Confidence:** 3

**Summary:**

This paper introduces **Meta-RFFT**, a length-aware SFT approach that explicitly teaches the model to follow step-by-step rules. After mastering these rules, the model generalises to longer and more complex scenarios, and experiments show strong transfer to unseen tasks.

**Questions:**

1. Will the authors release the codebase?
I believe the task construction pipeline is highly valuable for the community. Releasing the codebase would greatly facilitate future research and adoption of the proposed benchmark.

2. Can the proposed approach be extended to reinforcement learning frameworks?
Reinforcement learning is often considered to offer better generalization capabilities. It would be interesting to see whether this dataset can be adapted for RL settings, and whether RL-based models demonstrate stronger compositional generalization across tasks. I would be particularly curious to compare the performance of RL models on such benchmarks.

3. Can we authors do some ablation study on evaluating how performance scales with number of composed sub-tasks?
I believe not all subtasks contribute equally to performance gains. A systematic analysis of how accuracy changes, whether it degrades or holds up as the number of composed tasks increases from two to three or four would further support the paper’s claims about compositional generalization.

**Ethical Concerns:**

["NO or VERY MINOR ethics concerns only"]

**Limitations:**

Yes.

**Paper Formatting Concerns:**

No.

**Quality:**

3

**Strengths And Weaknesses:**

**Strengths**

1. The paper presents a thorough experimental study across a diverse set of tasks and model scales. The consistently strong results highlight the promise of the proposed method.
2. The authors’ data-synthesis and evaluation pipeline is itself a valuable asset. Open-sourcing this codebase would greatly benefit the community.
3. The proposed method for enhancing meta-thinking in reasoning tasks is both insightful and valuable.

**Weakness / Future Work**

* The current approach is framed within supervised fine-tuning (SFT). Because reinforcement learning (RL) can naturally capture meta-information during training, it would be interesting to see whether the method can be generalized to RL in future work.

---

> ### Author Rebuttal · Authors · 2025-07-31
>
> We sincerely appreciate the reviewer’s insightful and constructive feedback! We hope the following responses will further address your concerns.
>
> # W1, Q2:
> We find the reviewer’s suggestion to explore RL for rule-following tasks highly insightful, and in fact, we have conducted preliminary investigations along this line.
>
> In our current work, we explore a baseline using RL **with outcome rewards** through experiments. While outcome-reward RL has achieved success in general reasoning, we find it **less suited for strict rule-following**. Unlike supervised fine-tuning (SFT), which directly maximizes the likelihood of generating rule-compliant intermediate steps, outcome-based RL optimizes only for the final answer. For tasks with **explicit human-defined rules**, we believe **explicitly teaching the rule** is more effective than relying on model's exploration.
>
> Empirically, as shown in Figure 7, the RL-trained model underperforms Meta-RFFT. Without supervision of intermediate steps, it struggles to follow rules reliably (see Table 9 for an error case). Besides, Meta-RFFT significantly outperforms long-CoT models like DeepSeek-R1-671B and QwQ-32B on length generalization tasks (see R1's error case in Table 8).
>
> That said, we strongly agree with the reviewer tbVk (and reviewer VnFA) that **RL, especially multi-task RL with process-level rewards** is a promising future direction.
>
> # Q1:
> Yes, we appreciate your interest! The data construction code is already included in the supplementary material, and we will release the full codebase, including task generation scripts and training pipelines, to support future research and adoption.
>
> # Q3:
> We appreciate your insightful suggestion and have conducted the following ablation study.
>
> In the current paper, the RF-pretraining stage includes tasks from three datasets: LeetCode, Big-Bench Hard, and Symbolic Reasoning. To assess the impact of the number of composed tasks, we conduct an ablation where the model is RF-pretrained **only on the LeetCode subset**.
>
> Due to resource constraints, we evaluate on 4 downstream tasks, including 2 LeetCode tasks (Move Zeros & Valid Palindrome) and 2 NUPA tasks (Add Integer & Digit Max Integer), all of which are unseen during RF-pretraining.
>
> The results are shown below. We observe **a slight performance drop** when excluding Big-Bench Hard and Symbolic Reasoning, likely due to reduced task diversity during pretraining:
>
> |                             | ACC_Len30 | Max_Len_90% |
> |-----------------------------|-----------|-------------|
> | Meta-RFFT (complete pretrain dataset)                 | 0.56      | 16.0       |
> | Meta-RFFT (LeetCode only)   | 0.51      | 12.5       |

---

> > ### Comment · Reviewer_tbVk · 2025-08-04
> >
> > Thanks for the clarifications and the additional experiments.
> >
> > Regarding the RL component, I think a format-reward (a simple reward that explicitly penalises outputs violating the prescribed format) may could help model follow the format.
> >
> > I still find the work compelling. The promised release of the codebase and dataset will be a valuable contribution to the community. Therefore, I maintain my score of 5 and recommend acceptance, while urging the authors to include the format-reward experiment in the final version.

---

> > > ### Author Response · Authors · 2025-08-04
> > > **Thank you for your positive feedback!**
> > >
> > > Thank you for your positive feedback and constructive suggestions! We appreciate that you find our work compelling.
> > >
> > > Your suggestion about incorporating a format-reward to explicitly penalize format violations is highly insightful. We agree that this could further improve the model’s adherence to the prescribed output structure, and we will conduct additional experiments to explore this idea in the final version.
> > >
> > > As promised, we will release the codebase and dataset to facilitate future research.

---

### Official Review · Reviewer_AGZE · 2025-07-02

**Clarity:** 3
**Significance:** 2
**Originality:** 3
**Rating:** 4
**Confidence:** 4

**Summary:**

The paper introduces Meta Rule-Following Fine-Tuning (Meta-RFFT), a framework to enhance length generalization in large language models (LLMs). Unlike prior single-task approaches, Meta-RFFT fine-tunes models on a diverse dataset of 86 tasks (spanning code execution, number processing, and logical reasoning), enabling cross-task transfer of rule-following skills. Key contributions include constructing a 310k-sample dataset, demonstrating that Meta-RFFT outperforms state-of-the-art models (e.g., 95% accuracy on 30-digit addition vs. 72% for DeepSeek-R1-671B), and showing that models learn shared computational primitives (e.g., loop maintenance) rather than task-specific patterns. The framework supports one-shot prompting and adapts to natural language rules, reducing the need for extensive task-specific training.

**Questions:**

NA

**Ethical Concerns:**

["NO or VERY MINOR ethics concerns only"]

**Final Justification:**

I was mainly concerned about the method's generalization to real-world tasks and its high computational cost. The author's rebuttal resolves most of my concern and thus I raise my score.

**Limitations:**

Yes.

**Quality:**

3

**Strengths And Weaknesses:**

Strength:

This work makes clear contributions against previous work:
(1) Meta-RFFT addresses a critical gap in prior work by enabling LLMs to generalize across tasks with minimal fine-tuning, the model achieves robust performance on unseen lengths by leveraging shared reasoning patterns.
(2) Extensive experiments across model sizes (7B–32B) and tasks (e.g., LeetCode, NUPA) validate its superiority, error analysis shows Meta-RFFT reduces loop-related errors by 50% compared to single-task methods, confirming its focus on generalizable reasoning.
(3) The large, diverse dataset and automated rule-trajectory generation streamline training, while the framework’s compatibility with practical LLMs (e.g., Qwen-2.5) enhances real-world applicability.

Weakness:
Even though this work has clear strengths, I have several concerns.

1. Limited Real-World OOD Generalization and Evaluation
The framework primarily evaluates structured tasks (e.g., arithmetic, code execution) and lacks rigorous testing on unstructured, real-world scenarios like article writing, deep research tasks, etc. The dataset requires the question to be answered by specific rules simplifies the creation but also increase the gap between these synthetic tasks and real-world applications. I think the LLM developers may not be very interested in improving length generalization on digit addition.

2. High Computational Overhead for Multi-Task Pretraining
Meta-RFFT requires substantial compute resources for RF-pretraining: 144–713.6 GPU hours for 7B–32B models, which may be inaccessible to researchers with limited infrastructure. While downstream fine-tuning is efficient (2.4–12 GPU hours per task), the upfront cost of multi-task pretraining poses a barrier to adoption for small teams or academic labs .

3. While the paper positions Meta-RFFT as a solution for length generalization, its core contribution—multi-task rule transfer—seems a bit off from the length generalization focus.

---

> ### Author Rebuttal · Authors · 2025-07-31
>
> We sincerely thank the reviewer for recognizing our key contributions, including **cross-task generalization**, **strong empirical results**, and **practical scalability**. We hope the following responses help address your remaining concerns.
>
> # W1:
> This is a valid and important concern. While our current evaluation focuses on structured tasks, the goal is to establish a foundation for robust generalization that can eventually transfer to more complex and unstructured real-world settings. Below, we explain the rationale behind our task choices and how they connect to broader applications.
>
> 1. **Additional experiments on more complex tasks**
>     * To validate the practical value of Meta-RFFT beyond symbolic tasks, we additionally conduct experiments and evaluate it on a real-world scenario: **the Airline Baggage Cost Estimation task** from [1].
>
>     * This task asks the model to compute the total travel cost for one or more passengers, including flight fare and baggage fees. The rules which are based on American Airlines' policy are complex, depending on cabin class, flight route, number of bags, and bag dimensions. Solving it **requires understanding and applying natural language regulations, which go beyond what can be fully captured in formal code**. We use pseudo-code rules in this task and test whether the model can perform length generalization.
>
>     * This task illustrates a practical setting for length generalization, where the model must apply multiple interrelated rules over long reasoning chains. Current LLMs perform poorly on it, and even strong models like Claude-3.5 and o1-preview degrade rapidly with input length (see the table below).
>
>     * We train models on samples with length 1–8, and test on OOD lengths 10–16. As shown below, **Meta-RFFT significantly outperforms vanilla RFFT in the practical domain**:
>
>     | length       | 2    | 4    | 6    | 8    | 10   | 12   | 14   | 16   |
>     |--------------|------|------|------|------|------|------|------|------|
>     | Vanilla RFFT | 0.94 | 0.72 | 0.78 | 0.56 | 0.44 | 0.16 | 0.00 | 0.08 |
>     | Meta-RFFT    | 1.00 | 1.00 | 0.92 | 0.90 | 0.74 | 0.38 | 0.24 | 0.26 |
>
>     | length                     | 5    | 8    | 11   |
>     |----------------------------|------|------|------|
>     | Qwen-2.5-72B (0-shot)      | 0.01 | 0.01 | 0.00 |
>     | Qwen-2.5-72B (1-shot)      | 0.19 | 0.10 | 0.01 |
>     | Claude-3.5 Sonnet (0-shot) | 0.04 | 0.00 | 0.01 |
>     | Claude-3.5 Sonnet (1-shot) | 0.29 | 0.30 | 0.11 |
>     | o1-preview (0-shot)        | 0.54 | 0.37 | 0.21 |
>     | o1-preview (1-shot)        | 0.63 | 0.55 | 0.46 |
>
>
> 2. **Why we choose symbolic tasks in our paper**
>     * We agree that extending our method to real-world, complex tasks is a valuable future direction. In this paper, we intentionally focus on controlled tasks to clearly isolate the challenges of length generalization, independent of task complexity of each step. This setup follows the standard in prior studies [2–5]. Notably, even when individual steps are simple, length generalization remains highly challenging for LLMs. For example, DeepSeek-R1-671B achieves only 72% accuracy on 30-digit addition, despite the task’s apparent simplicity to humans. This demonstrates that **length generalization is a fundamental and non-trivial ability, separate from broader reasoning complexity**.
>     * Furthermore, we also acknowledge the challenge of defining "length" in unstructured real-world tasks. Extending our framework to such domains is a key direction we plan to pursue.
> 3. **Our contribution towards real-world settings**
>     * Our work bridges the gap between prior small-scale, single-task studies and practical LLMs. We show that multi-task rule-following training enables transferable length generalization, including to unseen tasks. This provides a scalable path toward improving reliability in multi-step, real-world reasoning.
> # W2:
> Thank you for highlighting this important practical consideration. We’d like to address your concern as follows:
>
> 1. **Reducing pretraining cost with fewer steps:**
>    As shown in Figure 15 (Appendix E.5), the model’s performance improves steadily with more RF-pretraining steps, but even **partial pretraining (e.g., fewer steps)** yields notable gains over vanilla RFFT. This provides a **flexible trade-off** for users with limited resources: by reducing pretraining steps, one can still achieve meaningful improvements in length generalization.
>
> 2. **Cost amortization across tasks:**
>    While Meta-RFFT incurs an upfront cost (144 GPU-hours for 7B; 713.6 for 32B), it becomes **significantly more efficient than vanilla RFFT** when applied across multiple tasks. Since Meta-RFFT enables **positive transfer via 1-shot prompting** on **unseen tasks**, it reduces the need for downstream fine-tuning altogether. For example, with downstream adaptation costing just 2.4–4.4 GPU-hours per task (7B), the pretraining cost is amortized after only **\~42 tasks**—a threshold quickly exceeded in practice. For the 32B model, the breakeven point is **\~72 tasks**. This makes Meta-RFFT cost-effective for large-scale multi-task deployment. We will also open-source the Meta-RFFT checkpoints for efficient downstream applications.
>
> |Model|Training Stage|Training hours|GPU Num|GPU Memory|GPU hours|
> |-|-|-|-|-|-|
> |7B|RF-pretrain|18|8|80G|144.0|
> |7B|downstream|0.6~1.1|4|80G|2.4~4.4|
> |32B|RF-pretrain|22.3|32|80G|713.6|
> |32B|downstream|2.0~3.0|4|80G|8.0~12.0|
> # W3:
> We would like to clarify how **multi-task rule transfer and length generalization are closely connected** in our work.
>
> As identified in [5], poor length generalization in LLMs stems from reliance on **case-based reasoning**, where models memorize shallow patterns instead of learning genuine rules. For example, models memorize addition results within 5-digit numbers by seeing a large number of training examples in this range, but do not capture the underlying digit-by-digit with carry-over addition rules (which are the key for generalization to infinite lengths), thus fail to solve 10-digit addition problems. RFFT addresses this by providing explicit rules and execution traces, enabling the model to adopt **rule-based reasoning**, which significantly improves out-of-distribution (OOD) generalization.
>
> Our key contribution builds on this by **scaling rule-based reasoning to the multi-task setting**. By training on a large number of (rule, execution trace) pairs from diverse tasks, Meta-RFFT equips the model with a **meta rule-following ability**—allowing it to **transfer rule-following behavior across tasks**. Crucially, this enables the model to generalize not only to longer inputs but also to **unseen tasks**.
>
> Thus, while Meta-RFFT introduces multi-task rule transfer, its primary goal remains to improve **length generalization in a more generalizable and scalable way**. We also provide a mechanistic explanation: this transferability emerges from learning **shared computational primitives** (e.g., loop maintenance), which are essential for both length generalization and cross-task generalization.
>
> In this sense, multi-task rule transfer is not a departure from our focus on length generalization, but a natural and scalable extension of it.
>
> ---
> [1] Zhou et al., RULEARENA: A Benchmark for Rule-Guided Reasoning with LLMs in Real-World Scenarios, 2025.
>
> [2] Anil et al., Exploring length generalization in large language models, 2022.
>
> [3] Zhou et al., What algorithms can transformers learn?, 2023.
>
> [4] Zhou et al., Transformers can achieve length generalization but not robustly, 2024.
>
> [5] Hu et al., Case-Based or Rule-Based: How Do Transformers Do the Math?, 2024.

---

> > ### Comment · Reviewer_AGZE · 2025-08-05
> >
> > Thanks for your reply, which resolves most of my concerns. I will raise my score.

---

> > > ### Author Response · Authors · 2025-08-06
> > > **Thanks for your positive feedback!**
> > >
> > > Thank you for your positive feedback! We sincerely appreciate your consideration and are grateful for your decision to raise the score. We will ensure that the additional results are included in the revised version of the paper.

---

### Official Review · Reviewer_5dGj · 2025-07-02

**Clarity:** 4
**Significance:** 3
**Originality:** 3
**Rating:** 5
**Confidence:** 4

**Summary:**

The authors aim to improve the model’s generalization ability on downstream tasks by leveraging meta-learning over a carefully constructed dataset of more than 80 reasoning tasks. This approach not only enhances generalization but also alleviates the burden of understanding complex rules during downstream fine-tuning. By improving upon the original RFFT framework, the proposed method achieves better performance in rule-based learning.

**Questions:**

1. It is unclear how the relative error reported in Figure 4 is computed. A more precise definition or formula would help in understanding the performance implications.
2. I would like to see results on more complex algorithmic problems, such as shortest path computation, or tasks that are difficult to describe using simple while loops. This would provide stronger evidence for the model’s generalization ability.
3. In the task design, can the length parameter be interpreted as the input size n in algorithmic complexity analysis? If so, is it appropriate to mix linear-complexity and quadratic-complexity tasks with the same length value during the first-stage training? This could potentially bias the model’s learning.

If these concerns can be addressed—particularly through clarification and additional experiments—I would be willing to consider raising my score.

**Ethical Concerns:**

["NO or VERY MINOR ethics concerns only"]

**Final Justification:**

The authors' rebuttal satisfactorily addressed most of my initial concerns, especially regarding the definition of the relative error and  more estimation task in reality. I have updated my score accordingly. However, I still find the approach based on mixed complexity somewhat problematic, as it introduces ambiguity in interpreting the model's performance.

**Limitations:**

1.The proposed method demonstrates strong performance on tasks that can be decomposed into multiple similar iterations. However, it lacks discussion or evaluation on reasoning problems beyond this class, such as tasks with non-iterative or structurally diverse reasoning patterns.
2.The paper lacks a more detailed explanation of how length is defined across different tasks, which makes it difficult to interpret the consistency and generality of the length generalization results.

**Paper Formatting Concerns:**

No major formatting concerns. The paper follows the NeurIPS formatting guidelines.

**Quality:**

4

**Strengths And Weaknesses:**

Strengths：
1. The paper is clearly written and easy to follow, and the problem it addresses has strong practical significance.
2. The authors construct a complex and diverse multi-task dataset to enhance the model's generalization capabilities.
3. The experiments are extensive and well-designed, with thorough analysis and convincing empirical support for the conclusions.

Weaknesses：
1.In most of the training samples presented in the paper, the provided rules remain largely within the scope of “mathematical” or “formally codable” tasks. The study of reasoning capabilities in language models should not be limited to such types of problems.
2. The algorithmic tasks selected from LeetCode are relatively simple, and the evaluation lacks coverage of more complex or challenging tasks.

---

> ### Author Rebuttal · Authors · 2025-07-31
>
> We sincerely appreciate the reviewer’s insightful and constructive feedback! We hope our responses below address your concerns and help clarify our key contributions.
>
> # W1,2; Q2:
>
> We thank the reviewer for raising this important point. We agree that going beyond symbolic or formally codable tasks is crucial for evaluating real-world reasoning. Below, we clarify our task choices and how our work sets the stage for broader generalization.
>
> 1. **Additional experiments on more complex tasks**
>     * To validate the practical value of Meta-RFFT beyond symbolic tasks, we additionally conduct experiments and evaluate it on a real-world scenario: **the Airline Baggage Cost Estimation task** from [1].
>
>     * This task asks the model to compute the total travel cost for one or more passengers, including flight fare and baggage fees. The rules which are based on American Airlines' policy are complex, depending on cabin class, flight route, number of bags, and bag dimensions. Solving it **requires understanding and applying natural language regulations, which go beyond what can be fully captured in formal code**. We use pseudo-code rules in this task and test whether the model can perform length generalization.
>
>     * This task illustrates a practical setting for length generalization, where the model must apply multiple interrelated rules over long reasoning chains. Current LLMs perform poorly on it, and even strong models like Claude-3.5 and o1-preview degrade rapidly with input length (see the table below).
>
>     * We train models on samples with length 1–8, and report test accuracy on OOD lengths 10-16. As shown below, **Meta-RFFT significantly outperforms vanilla RFFT in the practical domain**:
>
>     | length       | 2    | 4    | 6    | 8    | 10   | 12   | 14   | 16   |
>     |--------------|------|------|------|------|------|------|------|------|
>     | Vanilla RFFT | 0.94 | 0.72 | 0.78 | 0.56 | 0.44 | 0.16 | 0.00 | 0.08 |
>     | Meta-RFFT    | 1.00 | 1.00 | 0.92 | 0.90 | 0.74 | 0.38 | 0.24 | 0.26 |
>
>     | length                     | 5    | 8    | 11   |
>     |----------------------------|------|------|------|
>     | Qwen-2.5-72B (0-shot)      | 0.01 | 0.01 | 0.00 |
>     | Qwen-2.5-72B (1-shot)      | 0.19 | 0.10 | 0.01 |
>     | Claude-3.5 Sonnet (0-shot) | 0.04 | 0.00 | 0.01 |
>     | Claude-3.5 Sonnet (1-shot) | 0.29 | 0.30 | 0.11 |
>     | o1-preview (0-shot)        | 0.54 | 0.37 | 0.21 |
>     | o1-preview (1-shot)        | 0.63 | 0.55 | 0.46 |
>
>
> 2. **Why we choose symbolic tasks in our paper**
>     * We agree that extending our method to real-world, complex tasks is a valuable future direction. In this paper, we intentionally focus on controlled tasks to clearly isolate the challenges of length generalization, independent of task complexity of each step. This setup follows the standard in prior studies [2–5]. Notably, even when individual steps are simple, length generalization remains highly challenging for LLMs. For example, DeepSeek-R1-671B achieves only 72% accuracy on 30-digit addition, despite the task’s apparent simplicity to humans. This demonstrates that **length generalization is a fundamental and non-trivial ability, separate from broader reasoning complexity**.
>     * Furthermore, defining "length" in complex real-world tasks is non-trivial. Our framework does not yet include large-scale real-world tasks, but we view this as a critical next step.
> 3. **Our contribution towards real-world settings**
>     * Our work bridges the gap between prior small-scale, single-task studies and practical LLMs. We demonstrate, for the first time, that multi-task rule-following training enables cross-task transfer of length generalization, marking an essential step toward scaling to more realistic, multi-step tasks.
>
> [1] Zhou et al., RULEARENA: A Benchmark for Rule-Guided Reasoning with LLMs in Real-World Scenarios, 2025.
>
> [2] Anil et al., Exploring length generalization in large language models, 2022.
>
> [3] Zhou et al., What algorithms can transformers learn?, 2023.
>
> [4] Zhou et al., Transformers can achieve length generalization but not robustly, 2024.
>
> [5] Hu et al., Case-Based or Rule-Based: How Do Transformers Do the Math?, 2024.
>
> ---
> # Q1:
> We apologize for the lack of clarity. In Figure 4(b), we report the **relative error in loop counts**, defined as:
>
> $$
> \text{relative-error} = \frac{\left|\text{ground-truth-loop-num} - \text{model-loop-num}\right|}{\text{ground-truth-loop-num}}
> $$
>
> For each task, the reported value is the **average relative error** computed over test cases of input lengths from 2 to 30 (in increments of 2), with **50 samples per length**.
>
> We will revise the text to include this definition for clarity.
>
> # Q3:
> Thank you for the insightful question.
>
> Yes, in our task design, the length parameter can be roughly interpreted as the input size $n$ in algorithmic complexity terms. For example, in the word sorting task, length corresponds to the number of words ($O(n^2)$), while in coin flip, it denotes the number of flips ($O(n)$).
>
> However, during training, we do not provide the model with explicit length values. As shown in the RFFT input format (e.g., Figure 1, right), the model is not required to be aware of the value of $n$. Instead, it learns to manage loop entry and exit based on conditions, not on fixed input lengths. This design encourages length-agnostic reasoning and prevents overfitting to specific lengths seen during training.
>
> Empirically, despite training on tasks of mixed complexity, the model generalizes well across downstream tasks of varying complexity, outperforming all baselines. This suggests that the mixed training does not bias the model in practice.

---

> > ### Comment · Reviewer_5dGj · 2025-08-06
> >
> > Your explanation has addressed most of my concerns, and I will increase my score accordingly. However, I still believe that the approach based on mixed complexity is somewhat inappropriate. I hope you can further investigate the effects of the length parameter, as it partially reflects the model's reasoning ability in algorithmic tasks.

---

> > > ### Author Response · Authors · 2025-08-07
> > > **Thanks for the insightful and responsible review!**
> > >
> > > Thank you for your insightful and responsible review throughout the process! We sincerely appreciate your positive feedback and the time you’ve invested in evaluating our work, as well as your decision to raise the score.
> > >
> > > Regarding the final concern, we acknowledge the importance of examining the impact of length parameter and computational complexity in algorithmic tasks. Your suggestion is inspiring, and we will further explore these aspects.

---

### Official Review · Reviewer_VnFA · 2025-07-06

**Clarity:** 4
**Significance:** 2
**Originality:** 3
**Rating:** 5
**Confidence:** 4

**Summary:**

This paper proposes Meta-RFFT, enhancing length generalization in LLMs. Compared to previous methods like positional encodings, direct answer, scratchpad fine-tuning and RFFT, Meta-RFFT focus on rule based reasoning not case based and multi tasks setting to validate the generalization.

The authors construct a dataset with 86 diverse tasks and use Meta-RFFT to fine-tune models, enabling them to generalize unseen tasks with minimal adaptation.(minimal fine-tuning or one-shot ICL).

Experiments show significant improvements over baselines (e.g. with a 32B model achieving 95% accuracy on 30-digit addition while training on 1-5-digit addition). The success is attributed to learning shared computational primitives across tasks(rule based reasoning not case-based). And models Meta-pretrained on code-based rules could adapt to natural language rules, demonstrating versatility.

**Questions:**

1.Following Weakness, the main computational primtive in this paper is loop maintenance? Any other primitives? If not, then I think this may not be an efficient way to learn rule-based reasoning, since Long-CoT models actually do a good job in Table 2.

2.Does META-RFFT fine-tuning affect the general capabilities of models, such as knowledge or mathematics?

3.I think reinforcement learning is actually very suitable for tasks in paper. Because it has clear state transitions and reward signals, maybe process supervision is more suitable for these tasks.

4.By the way, is it normal to have reference in the abstract?(line 9)

**Ethical Concerns:**

["NO or VERY MINOR ethics concerns only"]

**Final Justification:**

The author's reply addressed most of my concerns, especially the novelty of multi-task learning. And the authors also showed in rebuttal that Meta-RFFT can be used in applications beyond symbolic reasoning and the general capabilities of LLM can be maintained through training with a small amount of additional data. So I maintain the rating and raise my confidence score.

**Limitations:**

yes

**Quality:**

3

**Strengths And Weaknesses:**

Strengths:

1.This paper presents solid work and the writing is good for ideas and organizations that are easy to understand.

2.The experiment setup in paper is complete and fair, and there are a lot of training dynamic, result figures and tables to support the author's claims.

3.The results are significantly improved compared to the baselines.

Weaknesses:

1.One of my main concerns is novelty. In fact, the multi-task training to promote generalization has been deep explored in other domains.

2.Following Weakness 1, the approach in this paper is to apply multi-task learning on some specific tasks' length generalization problem (although the paper mentions four datasets, they are basically digital and symbolic processing. How about multi step math problems?)

---

> ### Author Rebuttal · Authors · 2025-07-31
>
> We sincerely appreciate the reviewer’s insightful and positive feedback! We hope the following responses will fully address your concerns and further clarify our contributions.
>
> # W1:
> We fully acknowledge that multi-task training is a well-established concept across many domains. However, our work introduces a novel perspective by being the first to explore **cross-task transfer of length generalization** through multi-task training. Length generalization presents a unique challenge due to its inherent out-of-distribution (OOD) nature—models are trained on short sequences but must generalize to much longer ones. We show through experiments that **traditional multi-task methods like instruction tuning (e.g., Tulu3 in Table 3) fail under such conditions**, since the reasoning steps required at test time go far beyond those seen during training.
>
> Besides, while the original RFFT framework showed that explicitly providing rules improves rule-based reasoning in single-task settings, our contribution is to **scale this approach to multi-task training** across a large number of (rule, rule-execution trace) pairs. This enables the model to acquire a **meta rule-following capability**, allowing it to generalize to **unseen tasks** by transferring reasoning skills across tasks. We are, to the best of our knowledge, the first to systematically study this form of **cross-task OOD generalization** in LLMs. We also provide a mechanistic explanation: the transferability stems from the model’s ability to learn shared **computational primitives**, such as loop maintenance, which are essential across tasks.
>
> Moreover, previous studies on length generalization [2–5] has **largely been restricted to single-task, small-scale** settings, often using transformers **trained from scratch**. However, a key strength of large language models lies in their ability to handle a wide range of diverse tasks within a single model. Our work aligns with this core advantage: we explore length generalization in a **multi-task, post-training setting, using pretrained LLMs**. Our method enables generalization across tasks and lengths, making it far more practical and scalable for real-world use.
> # W2:
> We thank the reviewer for raising this important point. We agree that going beyond symbolic or formally codable tasks is crucial for evaluating real-world reasoning.
>
> 1. **Additional experiments on more complex tasks**
>     * To validate the practical value of Meta-RFFT beyond symbolic tasks, we additionally conduct experiments and evaluate it on a real-world scenario: **the Airline Baggage Cost Estimation task** from [1].
>
>     * This task asks the model to compute the total travel cost for one or more passengers, including flight fare and baggage fees. The rules which are based on American Airlines' policy are complex, depending on cabin class, flight route, number of bags, and bag dimensions. Solving it **requires understanding and applying natural language regulations, which go beyond what can be fully captured in formal code**. We use pseudo-code rules in this task and test whether the model can perform length generalization.
>
>     * This task illustrates a practical setting for length generalization, where the model must apply multiple interrelated rules over long reasoning chains. Current LLMs perform poorly on it, and even strong models like Claude-3.5 and o1-preview degrade rapidly with input length (see the table below, the results are from [1]).
>
>     * We train models on samples with length 1–8, and report test accuracy on OOD lengths 10-16. As shown below, **Meta-RFFT significantly outperforms vanilla RFFT in the practical domain**:
>
>     | length       | 2    | 4    | 6    | 8    | 10   | 12   | 14   | 16   |
>     |--------------|------|------|------|------|------|------|------|------|
>     | Vanilla RFFT | 0.94 | 0.72 | 0.78 | 0.56 | 0.44 | 0.16 | 0.00 | 0.08 |
>     | Meta-RFFT    | 1.00 | 1.00 | 0.92 | 0.90 | 0.74 | 0.38 | 0.24 | 0.26 |
>
>     | length                     | 5    | 8    | 11   |
>     |----------------------------|------|------|------|
>     | Qwen-2.5-72B (0-shot)      | 0.01 | 0.01 | 0.00 |
>     | Qwen-2.5-72B (1-shot)      | 0.19 | 0.10 | 0.01 |
>     | Claude-3.5 Sonnet (0-shot) | 0.04 | 0.00 | 0.01 |
>     | Claude-3.5 Sonnet (1-shot) | 0.29 | 0.30 | 0.11 |
>     | o1-preview (0-shot)        | 0.54 | 0.37 | 0.21 |
>     | o1-preview (1-shot)        | 0.63 | 0.55 | 0.46 |
>
>
> 2. **Why we choose symbolic tasks in our paper**
>     * We agree that extending our method to real-world, complex tasks is a valuable future direction. In this paper, we intentionally focus on controlled tasks to clearly isolate the challenges of length generalization, independent of task complexity of each step. This setup follows the standard in prior studies [2–5]. Notably, even when individual steps are simple, length generalization remains highly challenging for LLMs. For example, DeepSeek-R1-671B achieves only 72% accuracy on 30-digit addition, despite the task’s apparent simplicity to humans. This demonstrates that length generalization is a fundamental and non-trivial ability, separate from broader reasoning complexity.
>     * Furthermore, defining "length" in complex real-world tasks is non-trivial. Our framework does not yet include large-scale real-world tasks, but we view this as a critical next step.
> 3. **Our contribution towards real-world settings**
>     * Our work bridges the gap between prior small-scale, single-task studies and practical LLMs. We demonstrate, for the first time, that multi-task rule-following training enables cross-task transfer of length generalization, marking an essential step toward scaling to more realistic, multi-step tasks.
>
> [1] Zhou et al., RULEARENA: A Benchmark for Rule-Guided Reasoning with LLMs in Real-World Scenarios, 2025.
>
> [2] Anil et al., Exploring length generalization in large language models, 2022.
>
> [3] Zhou et al., What algorithms can transformers learn?, 2023.
>
> [4] Zhou et al., Transformers can achieve length generalization but not robustly, 2024.
>
> [5] Hu et al., Case-Based or Rule-Based: How Do Transformers Do the Math?, 2024.
>
> ---
> # Q1,3:
> We find the reviewer’s suggestion to explore RL for rule-following tasks highly insightful, and in fact, we have conducted preliminary investigations along this line.
>
> 1. **On computational primitives:**
>    While loop maintenance is indeed a core challenge for length generalization, our model also learns other common primitives such as `if-else` logic and operations on `lists` and `strings`. These structures are broadly shared across tasks and essential for handling symbolic and algorithmic reasoning. Meta-RFFT leverages these shared patterns to improve cross-task generalization.
>
> 2. **On reinforcement learning:**
>    * **Outcome reward:**
>      We find SFT more effective than outcome-based RL for rule-following tasks. RL optimizes only for final correctness, while SFT directly trains the model to generate rule-compliant intermediate steps. As shown in Figure 7, RL fails to capture the process structure and underperforms Meta-RFFT (see error case of R1 in Table 9). Moreover, Meta-RFFT outperforms advanced long-CoT models like DeepSeek-R1-671B and QwQ-32B (Table 2), despite being smaller (see error case of R1 in Table 8).
>    * **Process reward:**
>      We strongly agree that this is a valuable idea. Supervising rule execution with process-level rewards could enhance improve rule adherence and generalization, and we consider it a promising direction for future research.
>
> # Q2:
> We appreciate the reviewer for pointing out this important and practical concern.
>
> Meta-RFFT involves substantial RF-pretraining, which indeed can affect the model’s general capabilities due to distribution shift, known as catastrophic forgetting. But typically this is easily addressed by mixing some pretraining corpus during fine-tuning.
>
> To verify this, we conduct a supplementary experiment. As we don't have access to the pretraining data, we mix GSM8K training data into downstream finetuning alongside the original meta-rule-following data to verify that the reasoning ability on GSM8K can be kept after RFFT. More specifically, for each training example in GSM8K, we use Qwen2.5-7B-Instruct to generate 10 candidate answers and filter the corrrect traces as the training set, forming a filtered GSM8K dataset of 7182 samples.
>
> The results on length generalization (averaged across 8 LeetCode tasks) and GSM8K test accuracy are shown below. We observe that merely mixing 7k samples (compared to 310k Meta-RFFT data) effectively recovers the model’s reasoning performance on GSM8K, while maintaining the strong multi-task length generalization ability. Notably, Meta-RFFT still significantly outperforms vanilla RFFT, demonstrating that **general capabilities can be preserved with minimal additional data**.
>
> |       | ACC_Len30 | Max_Len_90% |
> |---------|-----------|-------------|
> | vanilla RFFT                | 0.34      | 7.00        |
> | Meta-RFFT                   | 0.82      | 22.75       |
> | Meta-RFFT with general data | 0.79      | 21.25       |
>
> |        | GSM8K Test ACC |
> |------------------|----------|
> | Qwen2.5-7B-Instruct  |  87.95%   |
> | Qwen2.5-7B-Instruct (Meta-RFFT with general data) |           87.80%     |
> # Q4:
> While we have seen references included in some abstracts, we look into the convention and find that abstracts are generally expected to be self-contained and citation-free. We will revise our abstract accordingly to remove the reference.

---

> > ### Comment · Reviewer_VnFA · 2025-08-07
> >
> > The author's clear responses addressed most of my questions. I have no more questions. Thus, I maintain my score and improve my confidence score.

---

> > > ### Author Response · Authors · 2025-08-07
> > > **Thanks for the positive feedback!**
> > >
> > > Thank you for your positive feedback! We are pleased that our responses have addressed your concerns and helped clarify the contributions of our work. We greatly appreciate your time and effort in reviewing our paper and are grateful for your positive response and improved confidence in our approach.

---

### Author Response · Authors · 2025-08-06
**Thanks for All Reviewers' Feedback; Willing to Address Further Questions**

Dear Reviewers,

We would like to express our sincere gratitude for the valuable feedback provided by each of you. Below, we summarize the key contributions of our original submission and the additional experiments we conducted during the rebuttal phase to address the concerns raised. We hope that our efforts have sufficiently addressed your questions.

# Summary of Our Original Submission:
Overall Contributions:
1. **First work to explore cross-task length generalization:**
    Our work introduces a novel perspective by being the first to explore **cross-task transfer of length generalization** through multi-task training. Length generalization presents a unique challenge due to its inherent **out-of-distribution (OOD)** nature—models are trained on short sequences but must generalize to much longer ones.

2. **Large-scale length generalization dataset:**
    We collect 86 different length generalization tasks with 310k training samples from four task domains including code execution, number processing, logic and symbolic reasoning, which significantly broadens the previous length generalization tasks which mainly focus on addition, sorting or other basic operations.

3. **Meta-RFFT outperforms baselines significantly**:
    Meta-RFFT show significantly better performance than baselines (direct answer, CoT, vanilla RFFT). Specifically, a 32B model fine-tuned on 1-5-digit addition achieves 95% accuracy on 30-digit addition, vastly outperforming reasoning models of comparable or much larger parameter size (DeepSeek-R1-671B: 72%; QwQ-32B: 32%) and vanilla RFFT (40%). Notably, these models with rule-following pretraining can solve unseen tasks with high accuracy with **only 1-shot prompting**.

4. **Cross-task transfer comes from shared computational primitive:**
    We show that the foundation model robustly acquires shared computational primitives, which are critical for cross-task generalization. Loop maintenance is a primary error source. However, Meta-RFFT-ed models exhibit significantly more precise loop maintenance in downstream tasks.
# Additional Experiments During the Rebuttal Phase:
To further validate Meta-RFFT and address the reviewers' concerns, we conducted extensive additional experiments and provided detailed explanations. These included:
1. **How does Meta-RFFT perform on real-world tasks? (*raised by Reviewer Reviewer VnFA, 5dGj, AGZE*)**:
    * To validate the practical value of Meta-RFFT beyond symbolic tasks, we additionally conduct experiments and evaluate it on a real-world scenario: the **Airline Baggage Cost Estimation task** from [1]. This task illustrates a practical setting for length generalization, where the model must apply multiple interrelated rules over long reasoning chains. Current LLMs perform poorly on it, and even strong models like Claude-3.5 and o1-preview degrade rapidly with input length. We show through experiments that **Meta-RFFT significantly outperforms vanilla RFFT in the practical domain**.
2. **How does Meta-RFFT affect general reasoning performance? (*raised by Reviewer VnFA, MTxe*)**:
    * Meta-RFFT involves substantial RF-pretraining, which indeed can affect the model’s general capabilities due to distribution shift, known as catastrophic forgetting. But typically this is **easily addressed by mixing some pretraining corpus during fine-tuning**.
    * To verify this, we conduct a supplementary experiment: We mix GSM8K training data into downstream finetuning alongside the original downstream rule-following data. We observe that merely mixing 7k samples (compared to 310k Meta-RFFT data) effectively **recovers the model’s reasoning performance on GSM8K, while maintaining the strong multi-task length generalization ability**. Notably, Meta-RFFT still significantly outperforms vanilla RFFT, demonstrating that **general capabilities can be preserved with minimal additional data**.
3. **Ablations of tasks in RF-pretraining (*raised by Reviewer tbVk*)**:
    * In the current paper, the RF-pretraining stage includes tasks from three datasets: LeetCode, Big-Bench Hard, and Symbolic Reasoning. To assess the impact of the number of composed tasks, **we conduct an ablation where the model is RF-pretrained only on the LeetCode subset**.
    *  We observe **a slight performance drop** when excluding Big-Bench Hard and Symbolic Reasoning, likely due to reduced task diversity during pretraining.
---
[1] Zhou et al., RULEARENA: A Benchmark for Rule-Guided Reasoning with LLMs in Real-World Scenarios, 2025.

# Reviewer-Author Discussion Update:
As the Reviewer-Author Discussion period is now halfway through, we are eager to continue addressing any additional questions or concerns you may have.

We sincerely appreciate your attention to this matter and look forward to hearing whether our rebuttal has sufficiently addressed your concerns, or if you are willing to revise your scores.

---

### Decision · Program_Chairs · 2025-09-17

**Decision:**

Accept (poster)

**Comment:**

This paper presents work on length generalization in large language models. It introduces Meta-RFFT, a multi-task fine-tuning framework that enables robust cross-task generalization. A new dataset of 86 tasks, spanning symbolic reasoning, code execution, and number processing, has been curated. Experiments also demonstrate strong performance gain.

The work is well motivated, with comprehensive empirical results. It receives the comments of five reviewers. During the review process, reviewers raised concerns in (1) limited task types; (2) mixed complexity in training; (3) the framework overhead; (4) the extension to reinforcement learning strategy. A detailed rebuttal is provided to address them. Four of five reviewers are finally positive about this work. One reviewer is a bit negative, mainly due to the concern about the technical contribution of this work.

AC has carefully reviewed the paper and the reviewers’ comments, and considers that this paper can provide enough valuable insights to the research field and meets the acceptance line. The authors are encouraged to incorporate the reviewers’ constructive feedback in the final version to further strengthen the clarity and impact of this work.